# Wi-GC: A Deep Spatiotemporal Gesture Recognition Method Based on Wi-Fi Signal

Xiaochao Dang [1,2,*], Yanhong Bai [1] 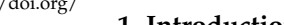, Zhanjun Hao [1,2] and Gaoyuan Liu [1]

1   College of Computer Science and Engineering, Northwest Normal University, Lanzhou 730070, China
2   Gansu Province Internet of Things Engineering Research Center, Lanzhou 730070, China
*   Correspondence: dangxc@nwnu.edu.cn

**Abstract:** Wireless sensing has been increasingly used in smart homes, human–computer interaction and other fields due to its comprehensive coverage, non-contact and absence of privacy leakage. However, most existing methods are based on the amplitude or phase of the Wi-Fi signal to recognize gestures, which provides insufficient recognition accuracy. To solve this problem, we have designed a deep spatiotemporal gesture recognition method based on Wi-Fi signals, namely Wi-GC. The gesture-sensitive antennas are selected first and the fixed antennas are denoised and smoothed using a combined filter. The consecutive gestures are then segmented using a time series difference algorithm. The segmented gesture data is fed into our proposed RAGRU model, where BAGRU extracts temporal features of Channel State Information (CSI) sequences and RNet18 extracts spatial features of CSI amplitudes. In addition, to pick out essential gesture features, we introduce an attention mechanism. Finally, the extracted spatial and temporal characteristics are fused and input into softmax for classification. We have extensively and thoroughly verified the Wi-GC method in a natural environment and the average gesture recognition rate of the Wi-GC way is between 92–95.6%, which has strong robustness.

**Keywords:** Wi-Fi; gesture recognition; channel state information; attention mechanism; RAGRU

## 1. Introduction

Gesture recognition plays a vital role in the research field of Human-Computer Interaction (HCI) [1]. It supports many emerging Internet of Things (IoT) applications such as user recognition [2], smart home [3], healthcare [4], etc. Generally, the technologies based on gesture recognition include sensors [5], web cameras [6], and millimeter-wave radars [7]. However, they all have certain limitations. For example, the sensor will cause an additional body burden to the user and its deployment and maintenance costs are high. The camera will expose the user's privacy and dead spots in the shot. Millimeter-wave radar signal attenuation is significant and the price is high. Recently, Wi-Fi-based gesture recognition methods [8] have become a hot research topic with non-contact, easy deployment, security and low-cost advantages. However, most of the current Wi-Fi-based gesture recognition methods extract the temporal features of gestures and ignore the spatial features, which affects the accuracy of gesture recognition to different degrees. To this end, we propose a method to obtain both temporal and spatial characteristics of gestures, thus improving gesture recognition accuracy in Wi-Fi-based environments.

Based on the limitations of the above gesture recognition technology, with the development of wireless communication and passive sensing technology, commercial Wi-Fi has become a research hotspot in the fields of gesture recognition [9], fall detection [10], breathing and heartbeat monitoring [11] and motion perception [12]. Previous researchers have mainly used Received Signal Strength (RSS) in Wi-Fi to sense gesture activity. For example, Sigg et al. used RSS generated by software radio to recognize gestures and achieved about 72% recognition accuracy [13]. Since RSS belongs to the Media Access Control (MAC)

layer, it is easily affected by path attenuation, occlusion and multi-path effects and has poor stability. However, CSI in Wi-Fi can compensate for RSS's shortcomings. Because it is a channel feature extracted from the PHY layer, it is a more detailed description of the channel and can also capture the multi-path variation of the signal propagation path. Therefore, CSI can clearly describe the influence of various human behaviors on signal propagation. For example, Thariq et al. [14] proposed a CSI-based gesture recognition system. The CIFE technique derives Bi-spectral Features (BF) from raw CSI data to explore Higher-Order Statistics (HOS) methods. The extracted features are finally classified with a Support Vector Machine (SVM) to form a subset of informative and optimal characteristics. Zhang et al. [15] proposed a dynamic gesture recognition algorithm based on CSI and You Only Look Once: Version 3 (YOLOv3). The data collection adopts the CSI-based radio frequency method. Grey value images are generated using adaptive weighted fusion, Kalman filtering, threshold segmentation and data transformation on the acquired data. Finally, the YOLOv3 object detection algorithm is used to train and recognize grayscale images containing continuous dynamic gesture information, with an average recognition accuracy of 94%. Therefore, compared with RSS, CSI is more stable, so the accuracy of gesture recognition is higher.

Currently, most CSI-based gesture recognition researchers extract gesture features from the amplitude or phase of CSI. For gesture recognition methods, extracting gesture features is a significant part. This method only focuses on some features of gestures and ignores others. Therefore, discovering other features of CSI data becomes a new challenge.

To solve the above problems, we propose a deep spatiotemporal gesture recognition method based on Wi-Fi signal (Wi-GC). The process mainly includes three stages. The first stage is data acquisition. The second stage is preprocessing: first, we select the antenna sensitive to gestures, then use the combined filter for noise reduction and smoothing, and then use the time series difference algorithm to split the data. The third stage is feature extraction and classification. We use BAGRU to extract temporal features of CSI and RNet18 to extract spatial features. Finally, their extracted features are fused and input into softmax for classification. The main contributions of this work are as follows:

- We have used the ubiquitous commercial Wi-Fi infrastructure to design a deep space-time gesture recognition method based on Wi-Fi signals. Due to low cost, no need to carry any equipment and no privacy leakage, it can be applied in many fields.
- We propose a feasible feature extraction method RAGRU, which can extract temporal and spatial features from CSI data. We add an attention mechanism to this algorithm, which can give more important weights to important gesture features to obtain high-precision classification results.
- We have conducted experiments in two natural environments on datasets of two environments collected by ourselves, verified the system's effectiveness for gesture recognition under Wi-Fi signals and evaluated the system's performance. The experimental results show that the average accuracy of the gestures was above 92% in both natural environments, which proves that the method has strong robustness and practicability in different settings.

The remainder of the article is organized as follows. Part 2 introduces the existing gesture recognition technologies and compares their advantages and disadvantages. Part 3 presents the specific process and algorithm of the Wi-GC method. Part 4 introduces the experimental scenarios and parameters and analyzes the impact of various factors on the experimental results used to evaluate the method's overall performance. Finally, Part 5 summarizes the full text and gives an outlook on future research work.

## 2. Related Work

### 2.1. Bound Gesture Recognition

At present, bound gesture recognition usually uses wearable sensors deployed in gloves, smart watches, etc. The commonly used sensors include accelerometer sensors, gyroscopes, inertial sensors, etc. Fang et al. [16] use inertial sensors in gloves to obtain

hand and arm data, then use the designed convolutional neural network structure SLRNet to extract the features of gesture data and classify them, with an accuracy rate of 99.2%. Zhu et al. [5] use accelerometers, linear accelerometers and gyroscope sensors in a smartwatch to collect gesture data, adopting the proposed gesture detection and segmentation algorithm to find out the start and finish points of the gesture and then converting the raw sensor reading data into Spectral features. Finally, a Bidirectional Long Short-Term Memory (BiLSTM) network is used for feature extraction and softmax classification. The gesture recognition rate is as high as 96%. Nguyen-Trong et al. [17] use accelerometers and gyroscopes in ordinary smartwatches to collect gesture data and combine a One-Dimensional Convolutional Neural Network (1D-CNN) with a BiLSTM network to analyze, learn and represent the sensory signals from the sensor signal features, reaching 90% accuracy. In addition, Lv et al. [18] use the somatosensory sensor to obtain the person's depth information and bone information, decompose the gesture into a gesture sequence composed of micro gestures, and then match the gesture sequence with the gesture template to obtain experimental results. The recognition rate reaches 92%. Jung et al. [19] proposed a method for muscle activity recognition, which detected muscle activity by measuring the air pressure change in the air sac contacting the muscle of interest, processed the data with high-pass and second-order low-pass filters and finally used the fuzzy logic method to classify gestures. Alfaro et al. [20] proposed a gesture recognition method using an inertial sensor and ElectroMyoGraphy (EMG) signal fusion and used fourth-order Butterworth high-pass filter to filter the signal for both EMG and Inertial Measurement Unit (IMU). The EMG is segmented by adopting the double-threshold technique. Each active segment of the EMG and IMU data is divided into 250 m overlapping windows and temporal features are obtained from the windows. Finally, an adaptive Least Squares Support Vector Machine (LS-SVM), a bilinear model-based classification method and a MultiLayer Perceptron (MLP) network for classification were utilized, with an average accuracy of 67.5–84.6%. Although bound gesture recognition has excellent advantages in recognition accuracy, it will cause an additional physical burden on users, affect people's lives and cause high deployment and maintenance costs.

### 2.2. Unbound Gesture Recognition

Four main types of unbound gesture recognition are currently studied: Radio Frequency Identification Devices (RFID), webcam, millimeter-wave radar, and Wi-Fi technology. RFID gesture recognition recognizes gestures through the signals received by RFID reading tags. For example, Cheng et al. [21] propose a real-time gesture recognition system, which collects gesture phase data reflected from passive RFID tags through a Commercial Off-The-Shelf (COTS) Impinj RFID reader. Using their own extracted algorithm to denoise and smooth the data with a Moving Average (MA) filter, they obtain static gesture features by relative position calculation and segmentally extract dynamic features. Finally, the K-Nearest Neighbours (KNN) and Dynamic Time Warping (DTW) algorithm are used to recognize gestures. The gesture accuracy rate reaches 90%. However, the RFID identification method requires gesture recognition under the condition of sight distance. Gesture recognition is based on a webcam process and recognizes the collected image data. For example, Nair et al. [6] firstly use the web camera to collect the picture data of gestures, thresholding to remove image noise and smooth the image, and finally use the KNN algorithm to extract and classify features, and the accuracy rate reaches 99.9%. Li et al. [22] present a novel skeleton-based dynamic gesture recognition framework. In a Spatially Perceptive stream (SP-stream), the compact joints are adaptively selected using their designed compact joint coding method for convex packages of the hand skeleton. They then encode them as skeleton images to fully extract spatial features. In addition, they provide a Global Enhancement Module (GEM) to enhance the critical feature maps. In the temporal perception stream (TP-stream), they propose a Motion Perception Module (MPM) to strengthen the significant motion of the gesture on the X/Y/Z axes. Then the Feature Aggregation Module (FAM) is used to aggregate more time dynamics. Finally, the

scores obtained from the spatial-aware and time-aware streams are averaged to get the final classification results. Verma et al. [23] proposed a hybrid deep-learning framework to recognize dynamic gestures. The features of each frame in the video need to be extracted to obtain temporal and dynamic information about the gestures made. Therefore, GoogleNet is used to extract the gesture features from the video. Finally, the extracted features are transferred to a Bidirectional Gated Recurrent Unit (BiGRU) network to classify gestures. Nguyen et al. [24] proposed a new continuous dynamic gesture recognition method. They use a gesture localization module to segment a video sequence of continuous gestures into individual gestures. Three residual 3D Convolution Neural Networks based on ResNet architectures (3D_ResNet) are used to extract the RGB, optical flow and depth of the gestures features. Meanwhile, BiLSTM is used to extract the features of the 3D positions of the critical joints of the gestures. Finally, the weights of the fully connected layers are fused for gesture classification. This method has a high recognition accuracy, but the method requires a massive amount of calculation and is easily affected by lighting conditions and obstacles. At the same time, the camera has a dead monitoring angle, which can only achieve perception within a specific range under the line of sight, and violates the user's privacy. Gesture recognition based on millimeter-wave radar uses millimeter-level signals sent and received by radar equipment and processes and recognizes them. For example, Zhang et al. [7] use millimeter-wave radar signals for gesture recognition and convert millimeter-wave data into a Time and Space Velocity (TSV) spectrogram. Then the gesture features are extracted by a specific feature extraction algorithm and classified with a custom classifier, achieving an accuracy of 93%. Although the gesture recognition of millimeter wave radar has high precision, its signal attenuation is significant and the cost is high.

The popularity and discovery of commercial Wi-Fi network infrastructure make up for the shortcomings of the above three technologies and provide solutions. Since CSI in Wi-Fi signals is fine-grained physical information, which comes from sub-carriers decoded in Orthogonal Frequency Division Multiplexing (OFDM) systems, it is susceptible to the environment. Therefore, it has received extensive attention from researchers in recent years. For example, Hao et al. [25] proposed a CSI-based sign language recognition method. This method uses the wavelet function to remove the noise in the environment. It uses the K-means combined with the Bagging algorithm to optimize the SVM classification model and the average recognition rate reaches 95.8%. Dang et al. [26] proposed a CSI-based aerial handwritten digit recognition system. First, the system selects the data that can reflect the gesture movement from the CSI raw data. At the same time, noise reduction processing is performed on the selected data. After processing, the amplitude and phase information features are extracted and the S-DTW algorithm matches and recognizes different air gestures. The average recognition accuracy of each action is over 93%. Han et al. [27] proposed a fine-grained gesture recognition method, denoising CSI amplitude with wavelet transform and use conjugate calibration to eliminate the phase shift of CSI. Then a Generative Adversarial Network (GAN) is used to enhance data, Deep Neural Network (DNN) to learn transferable features for domain adaptation, softmax to classify the extracted features, and gesture recognition. The average accuracy is 94.5%. Yang et al. [28] proposed a gesture recognition system based on Wi-Fi transmission physical layer CSI. First, the Savitzky-Golay filter is used to smooth the phase curve and amplitude of CSI, then 1D-CNN is used to extract gesture features, and finally, SVM is used for classification. The recognition rate of gestures is about 90%. Shi et al. [29] propose a gesture recognition method based on Wi-Fi signals. The amplitude and phase of the CSI are first extracted from the Wi-Fi signal, the amplitude is filtered and the phase is expanded and linearly transformed. The CSI signal is then converted to a Red Green Blue (RGB) image using normalization and interpolation. Finally, the combined amplitude and phase RGB images are classified using a lightweight deep network model based on MobileNet_V2. Wang et al. [30] propose a gesture recognition system based on a matched average joint learning framework (WiMA). The system uses parameter-matching collaborative learning to train a gesture prediction model. Static bias in the data is first removed using CSI amplitude

conjugate multiplication and noise and random bias are removed using a band-pass filter. The data's Doppler Frequency Shift (DFS) spectrum is extracted, and a Body-coordinate Velocity Profile (BVP) is generated. The features of the BVP are then extracted using CNN-LSTM and finally, the gesture features are classified using softmax. WiFi-Based Low-Complexity Gesture Recognition Using Categorization. Kim et al. [31] propose a low-complexity gesture classification recognition based on Wi-Fi. The CSI data is first processed using different techniques and the pre-processed data is segmented. Then features are extracted using a deep degree learning model and finally, the extracted features are classified using SVM. Ding et al. [32] proposed a gesture recognition scheme based on the multi-modal Gaussian mixture model (GMM). Their proposed GDS algorithm was used to segment the gesture data and then use Singular Value Decomposition (SVD) to derive multi-view features from CSI measurements on all subcarriers of the Wi-Fi receiver to represent gesture features, using a Multi-modal Factorized Bilinear (MFB) pooling method to efficiently fusion features from all receiver antennas, and finally identify various gestures by integrating multi-modal fusion and GMM.

### 3. Related Theory

#### 3.1. Basic Principles of Wi-Fi Human Perception

Wi-Fi signals will have multiple transmission paths in the process of propagation. The direct propagation path between the transmitter and the receiver is called a Line of Sight (LOS). When Wi-Fi signal propagation encounters obstacles such as ceiling and floor, reflection, scattering, etc., will occur. The propagation path is called Non-Line of Sight (NLOS). The influence model of human gestures on Wi-Fi signals is shown in Figure 1.

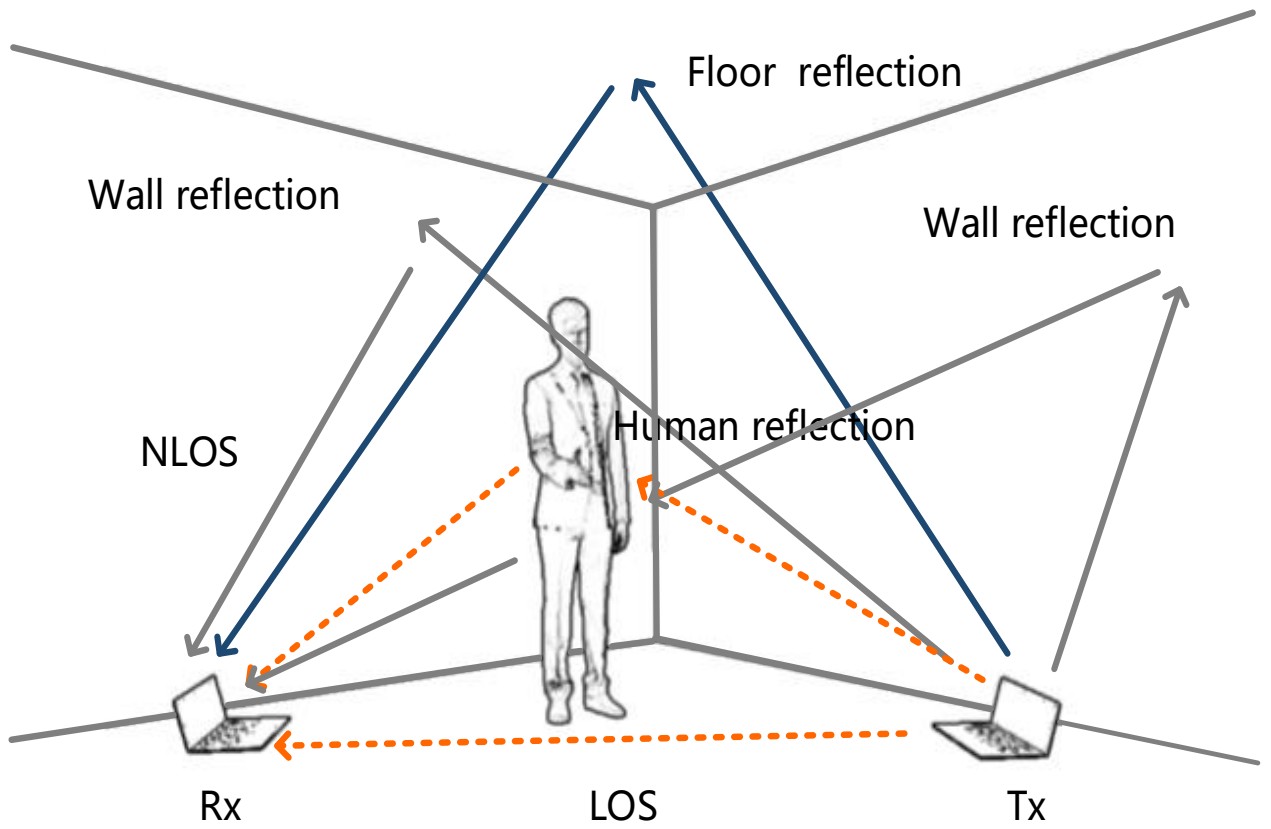

**Figure 1.** The effect of the human body on Wi-Fi signals.

The influence of human behaviour on Wi-Fi signal propagation will be presented in the Wi-Fi signal at the receiving end. Assuming that the LOS distance of the transceiver is $d$, the distance from the reflection point between the ceiling and the ground to the LOS is $h$, combined with the Friis free space propagation equation [33], the expression of the received power is as follows:

$$P_{ry}(d) = \frac{P_{ty}G_{ty}G_{ry}\lambda^2}{(4\pi)^2(d + 4h + \eta)^2} \tag{1}$$

In the formula, $d$ is the direct distance between the two ends of the transceiver. $P_{ty}$ is the transmit power of the transmitter. $P_{ry}(d)$ is the received power of the receiver. $G_{ty}$ is the transmit gain, $G_{ry}$ is the receive gain, $\lambda$ is the wavelength of the Wi-Fi signal and $\eta$ is approximated path length variation caused by human interference, due to the different scattering paths caused by various human behaviours to the propagating Wi-Fi signal. According to Equation (1), it can be seen that distinct human actions lead to differences in the received power at the receiver. These differences can be used to differentiate between different human behaviours.

*3.2. Channel State Information*

Wi-Fi perception aims to realize human behaviour, object and environment perception in the propagation space by analyzing the wireless signal propagation channel characteristics. The propagation characteristic of this wireless signal is CSI. In other words, CSI is the compensation of various channel effects at the receiving end, such as signal reflection, refraction, diffraction and multi-path attenuation, i.e., Channel Impulse Response (CIR). The channel frequency response (CFR) can be obtained using the fast Fourier transform (FFT). CFR can provide amplitude and phase information of subcarrier level through Multiple Input Multiple Output (MIMO) and OFDM technology. According to Ref. [9], CFR can be expressed as:

$$H(f,t) = \sum_{i=1}^{M} \alpha_i(f,t)e^{-j2\pi f\tau_i(f,t)} \tag{2}$$

where $\alpha_i(f,t)$ is the amplitude, $\tau_i(f,t)$ is the phase information and $M$ is the number of subcarriers.

During actual communication, Wi-Fi signals are subject to interference from hardware devices, the environment, etc., and the received CSI data contains associated noise. So the accepted CSI structure is usually expressed as:

$$H(f,t) = \left(\sum_{i=1}^{M} \alpha_i(f,t)e^{-j2\pi f\tau_i(f,t)}\right)e^{-j(2\pi t\Delta f + \theta_N + \theta_M)} \tag{3}$$

Formula $\Delta f$ represents the Central Frequency Offset (CFO) caused by the asynchronous clock between the transmitter and receiver; $\theta_N$ is the phase offset caused by the sampling frequency offset; $\theta_M$ is caused by hardware noise and the environment.

## 4. Method Design

The flow of our method is shown in Figure 2. First, we select the antenna sensitive to gestures, use the Kalman filter and Sym8 to denoise and smooth the selected antenna data and then use the time series difference segmentation algorithm to continuously segment gestures. The segmented data is inputted into the RAGRU model to extract features, and finally the extracted features inputted into softmax to obtain the classification result.

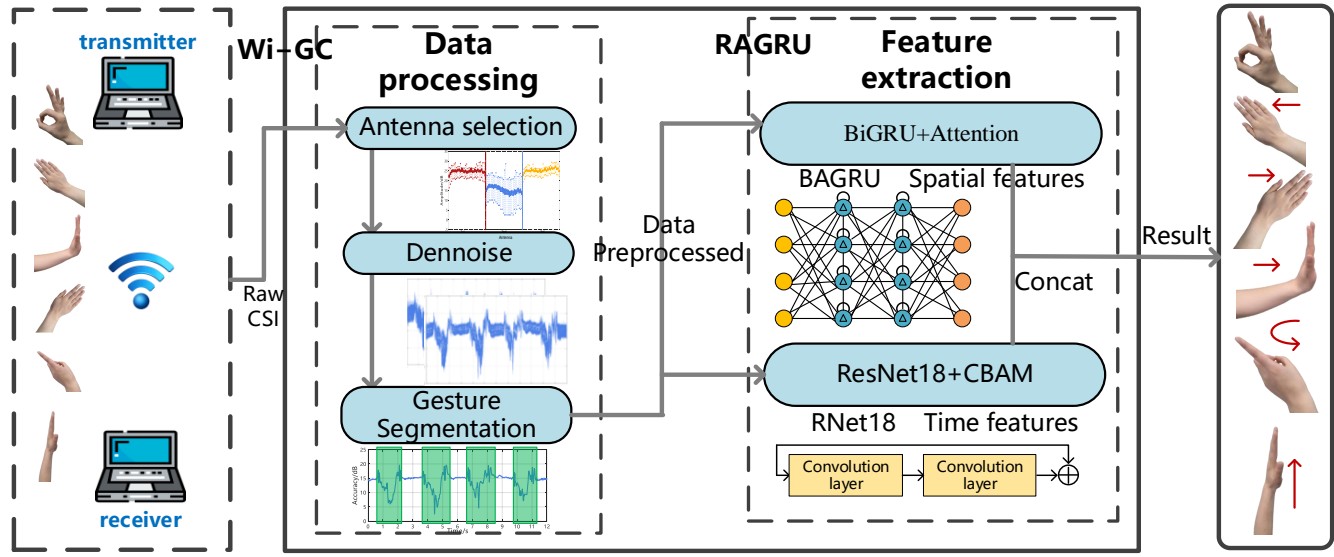

**Figure 2.** Flow chart of the Wi-GC method.

### 4.1. Data Collection

In the process of gesture data collection, we adopt the method of one transmission and three receptions, that is, one transmitting antenna and three receiving antennas, and each antenna can receive 30 subcarrier channel values. We designed six gestures, as shown in Figure 3. They are: "OK", "left arm slide", "right arm slide", "push", "rotate" and "raise your hand".

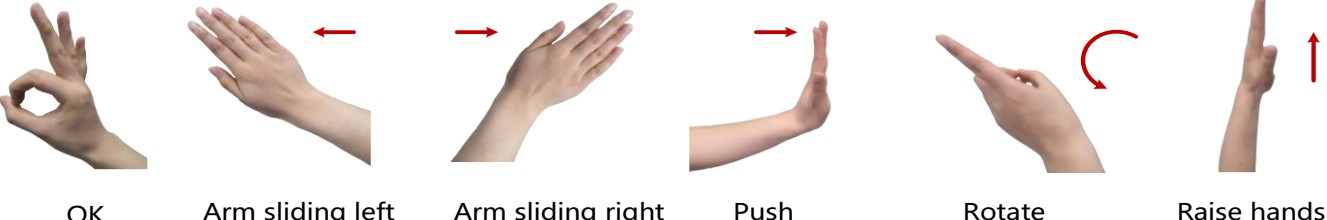

| OK | Arm sliding left | Arm sliding right | Push | Rotate | Raise hands |

**Figure 3.** Hand gestures.

### 4.2. Antenna Selection

Different antennas have different sensitivity to environmental perception. Extracting gesture features is not very sensible if the antenna is not sensitive to gestures. Therefore, we adopt the method of Ref. [34] to select the sensitive antenna. Figure 4a shows the amplitude distribution over the three antennas in a box plot. We observed that the second antenna, which has a lower average amplitude value, is more likely to produce a sizeable dynamic response. We observed that the second antenna with a low average amplitude value was likelier to have an enormous dynamic response. This means that the static component of the second antenna is weak and, therefore, very sensitive to small movements. Figure 4b shows the amplitude variation of subcarrier No. 1 on three different antennas. From its data, it is clear that the second antenna has enormous amplitude fluctuations compared to the first and third antennas. Inspired by the above two results, we choose the second antenna as a reference.

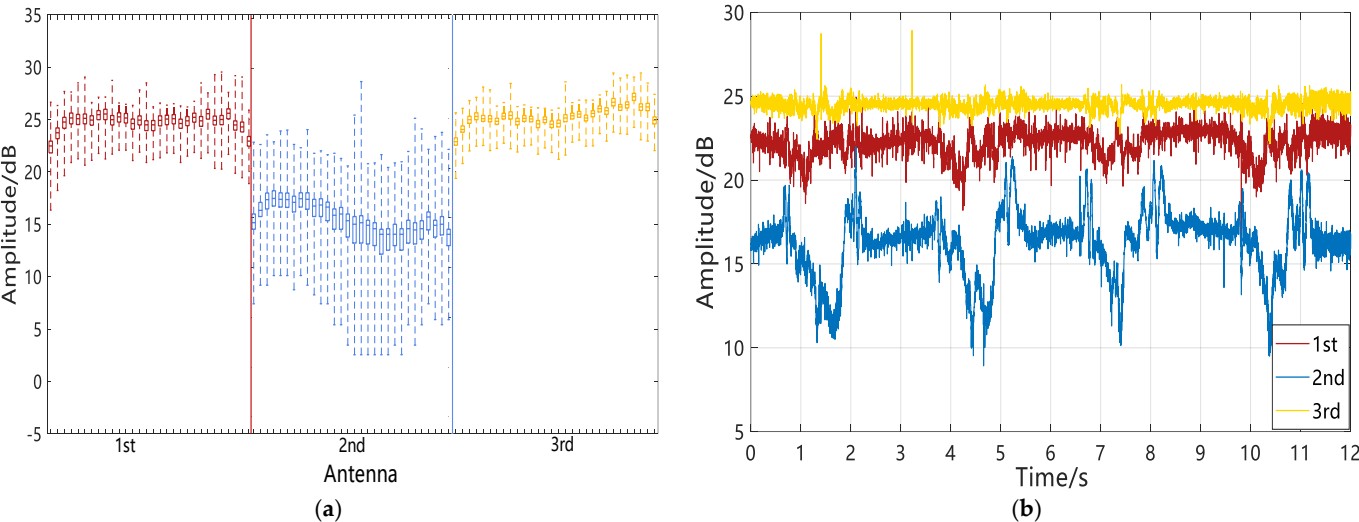

**Figure 4.** Antenna selection: (**a**) Comparison of the three antennas. (**b**) Subcarrier number one on three antennas.

### 4.3. Data Processing

Due to the significant environmental noise when collecting data, denoising is required. We chose the Kalman filter and Sym8 wavelet to process the noise of the collected raw CSI antenna data. The process of data processing is shown in Figure 5.

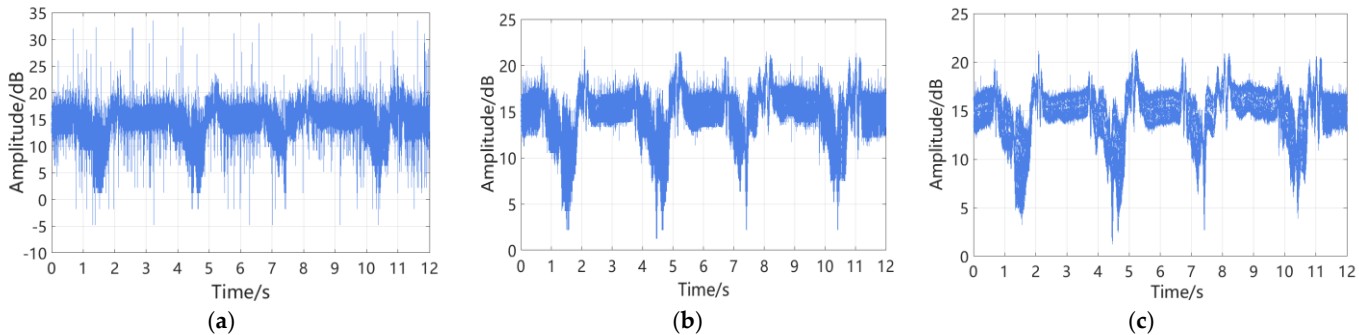

**Figure 5.** Data processing: (**a**) Raw data. (**b**) Data after Kalman filter processing. (**c**) Sym8 Wavelet processed data.

Figure 5a presents the raw antenna gesture data we have chosen, showing that it has many outliers. We use Kalman filtering to eliminate outliers away from the median or adjacent values of the original data [35]. This method mainly uses a moving average window to find outliers. It replaces them with the average value of the data, thereby eliminating the influence these outliers have on the data. The effect after processing is shown in Figure 5b, which shows that most outliers have been removed. Finally, Sym8 is used to remove the noise of the remaining part. The result is shown in Figure 5c. As we can see, the noise of the data has been completely removed and the data has become smooth.

### 4.4. Data Segmentation

We need to segment the collected continuous gesture data into individual gesture data. Therefore, we adopt the time series difference segmentation algorithm in Ref. [36], which in detail uses a series of overlapping sliding windows to calculate the average absolute difference of all CSI streams in each window.

$$\overline{Y} = \frac{\sum_{i=1}^{d} \sum_{j=s}^{s+w-1} |Y_{i,j+1} - Y_{i,j}|}{d \times w} \tag{4}$$

Among them, $d$ represents the CSI dimension, $s$ represents the starting point sum and $w$ represents the sliding window size. After that, the calculated value of $\overline{Y}$ is compared with the preset start threshold $T_1$ and end threshold $T_2$ ($T_1 > T_2$) to detect the gesture part

$$\begin{cases} P_s = j, \overline{Y} \geq T_1 \\ P_e = j, \overline{Y} \leq T_2 \text{ and } P_s \neq \varnothing \end{cases} \tag{5}$$

where $P_s$ is the start point of the gesture and $P_e$ is the endpoint. Finally, the false positive energies in the point set are removed by post-processing, since an antenna is composed of 30 subcarriers with similar waveforms, as shown in Figure 5. Therefore, we segment a subcarrier in the antenna, taking subcarrier number 15 as an example. The segmentation result is shown in Figure 6.

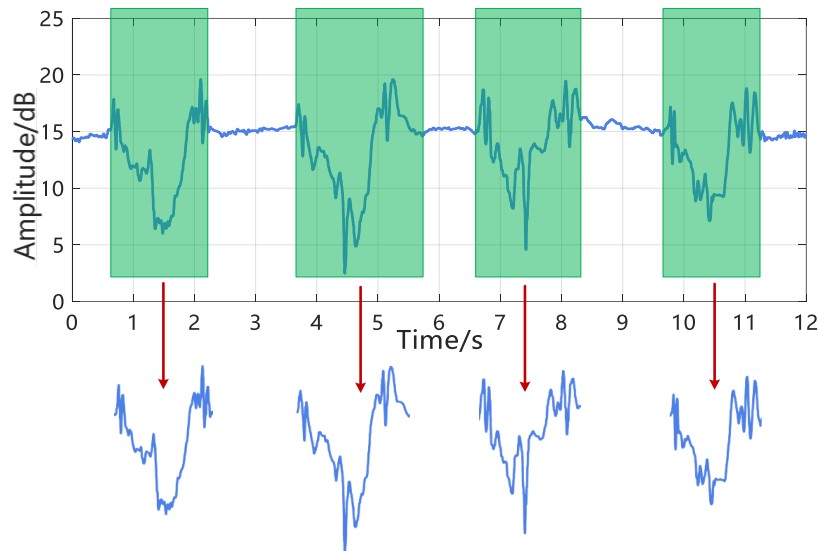

**Figure 6.** Gesture segmentation results.

*4.5. RAGRU Model*

Considering the spatial correlation of CSI amplitudes and the timing of CSI data [36], we propose to extract the spatial and temporal features of CSI data with the RAGRU model. The RAGRU model structure is shown in Figure 7.

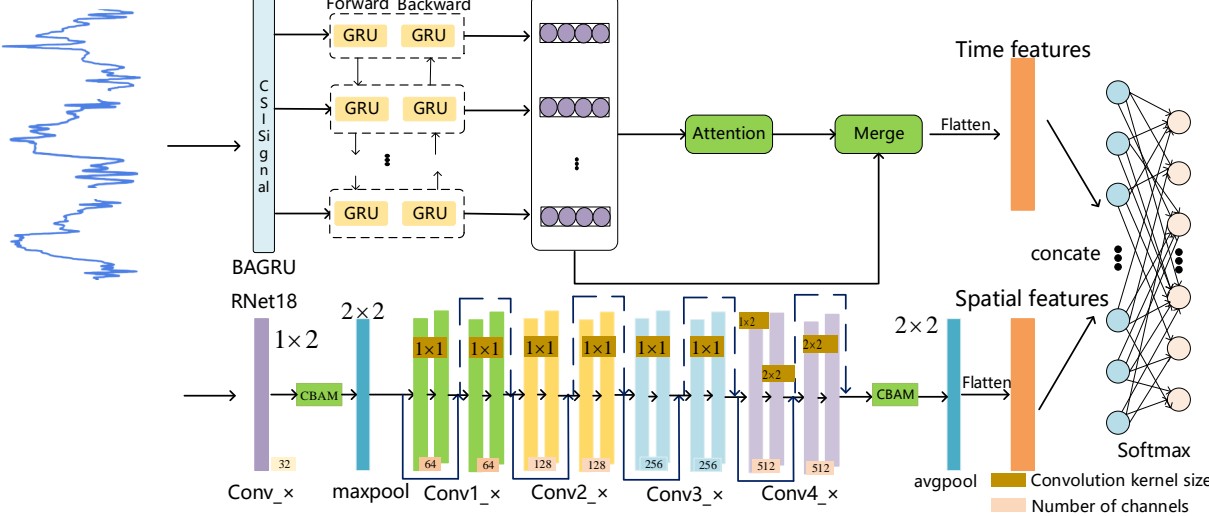

**Figure 7.** Flow chart of the RAGRU algorithm.

The RAGRU model feeds CSI stream data into the BAGRU and CSI amplitude data across the antenna into the RNet18 to obtain temporal and spatial features of gestures. Finally, the material features and space are fused and input into softmax for classification. Training of the proposed RAGRU model is shown in Algorithm 1.

---

**Algorithm 1.** Training with RAGRU

---

**Input:** Task set {$C_{train}$, $C_{test}$}, TC: target class, cw: class weight
**Output:** CR: Classification results
**while** *not done* **do**

1.  Sample a batch of tasks $TT_i \sim C_{train}$;
    **For** *eath* $TT_i$ **do**
2.  AGRU = BiGRU($TT_i$)
3.  $V_t$ = target_attention(AGRU, $TT_i$)//Calculate attention weight
4.  $V_s$ = attention(AGRU, $V_t$)//Assign attention weight to AGRU
5.  $h_{out}$ = $V_s \cdot W_{out} + b_{out}$
6.  S = Flatten($h_{out}$)
    **end for**
//BGRU training module on $C_{train}$;
    **For** *eath* $TT_i$ **do**
7.  F = Conv($TT_i$)
8.  $F' = M_C(F) \otimes F$
9.  $F'' = M_S(F') \otimes F'$
10. $F_L = F'' + \sum_{i=1}^{L-1} F(F_i'', W_i)$//Calculate CBAM
11. Excute steps 8, 9, get $F_L''$
12. $W = \text{avgpool}\left(F_L''\right)$
13. T = Flatten (W)
    **end for**
//RNet18 training module on $C_{train}$;
14. Fe = Concate (T + S)//Fusion features
15. output = softmax (Fe)
16. loss = compute_loss (output, TC, cw)
//Finetune model on Dtest;
17. Sample a batch of tasks $DD_i \sim C_{test}$
    **For** each $DD_i$ **do**
18. Excute steps 1~13
19. output = softmax(F)
20. loss = compute_loss (output, TC, cw)
21. CR = argmax(output)
    **end for**
**end for**

---

### 4.5.1. BAGRU

Due to the time-dependent capture and sequence modeling ability, Recurrent Neural Networks (RNNs) have been successfully applied in many fields [37]. However, initial RNNs do not make good use of historical information and, if the learned data sequence is too long, it will lead to vanishing gradients and significant problems [38]. The length of the learning sequence is generally positively related to the system's performance. In most cases, long learning sequences are necessary. A new RNN structure proposed in Ref. [39], namely LSTM, is proposed, which effectively compensates for the shortcomings of traditional RNNs by adding memory units of special gates. GRU is a very successful variant of LSTM. It was first presented in Ref. [40].

In our Wi-GC method, the BiGRU network is used to extract the timing features of gesture recognition. It differs from the traditional GRU network and the BiGRU has two layers: forward and backward. In the process of learning CSI information features, both past information and future information may be considered. Specifically, the forward layer encodes the information of the past time steps into the current hidden state, thus

taking into account the past data of the CSI sequence. The backward layer encodes the information of future time steps into the current hidden state, thus taking into account the future information of the CSI sequence. We use the BiGRU network, which learns all the information from the CSI data.

In addition, in the traditional BiGRU network, the learned features have the same weight in gesture classification. To resolve the problem, we introduce a self-attention mechanism in BiGRU, BAGRU, which can be used for each time step and feature, assign a weight and learn the importance of each time step and feature autonomously. Therefore, assigning higher weights to more important time steps and features improves method performance.

As shown in Figure 8, when $x_t$ and $h_{t-1}$ are input, the GRU at time $t$ updates itself in the following way.

$$z_t = \sigma(W_z \cdot [h_{t-1}, x_t]), \tag{6}$$

$$r_t = \sigma(W_r \cdot [h_{t-1}, x_t]), \tag{7}$$

$$\widetilde{h}_t = \tanh(W_h \cdot [r_t \times h_{t-1}, x_t] + b_h), \tag{8}$$

$$h_t = (1 - z_t) \times h_{t-1} + z_t \times \widetilde{h}_t, \tag{9}$$

where $w_z$, $w_r$ and $w_h$ are the weights and $b_h$ is the bias. The $\tanh(\cdot)$ function is the hyperbolic tangent and the $\sigma(\cdot)$ function is the Sigmoid activation function. $r_t$ is the reset gate, $z_t$ is the update gate, $\widetilde{h}_t$ is the candidate's hidden state and $h_t$ is the hidden state passed to the next moment. $[r_t \times h_{t-1}, x_t]$ represents the splicing of $r_t \times h_{t-1}$ and $x_t$. The $\tanh(\cdot)$ activation function constrains each element of $h_t$ to be within the range of $[-1, 1]$, while the Sigmoid activation function can assign each element a value between $[0, 1]$ to selectively forget or remember the current input, where 0 means forget all and 1 means all are reserved.

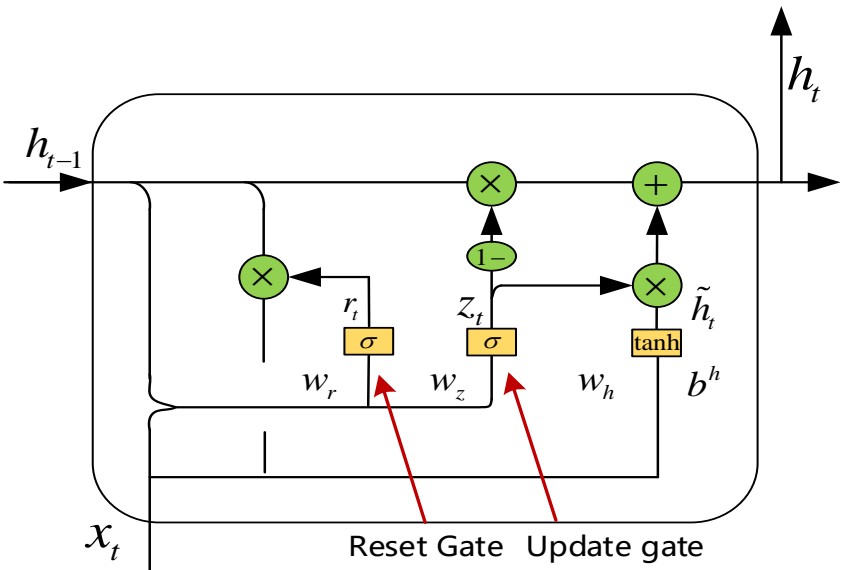

**Figure 8.** GRU structure diagram.

To consider both past and future CSI information, BAGRU consists of two layers, as shown in Figure 7. $h_f^t$ is the hidden forward state at time $t$ and $h_b^t$ is the hidden state of the backward layer at time $t$. Therefore, the hidden state of BAGRU can be represented as follows:

$$h_t = h_f^t \oplus h_b^t, \tag{10}$$

### 4.5.2. Attention Mechanisms

Daniel et al. [41] first proposed an attention model-based approach to image recognition inspired by the human visual system, claiming that humans always focus on a particular picture area in distinguishing the picture and adjusting the focus over time. With

the help of attention models, machines can focus only on the parts of interest and blur the rest simultaneously for recognition tasks, which is effective in image recognition and language processing tasks [42]. For example, in the machine translation of an article, the input sentence is encoded into a hidden vector with the same weight by the avant-garde encoder-decoder model. The translation is invalid because there is no attention mechanism. However, after adding the attention mechanism, the translation will pay more attention to the content related to the current word at different time steps to improve the translation performance. Despite this, BiGRU assigns equal weights to the time steps and features of all CSI data. Therefore, we introduce an attention mechanism to improve the recognition rate of gestures.

Since there is no prior information in CSI-based gesture recognition, the continuous sequence features learned by BAGRU are used as the input of the attention mechanism called self-attention [43]. For $m$ feature vector $f_j$, $j = 1, 2, \cdots m$, which can be derived from the feature learning network, we used a score function $\Phi(\cdot)$ to evaluate the importance of each feature vector by calculating the score $s_j$ as follows:

$$s_j = \Phi\left(W^s h^i + b^s\right), \tag{11}$$

Among these, $W^s$ is the weight and $b^s$ is the bias. Any activation function in the neural network can be used as a scoring function $\Phi(\cdot)$, such as Sigmoid, Rule, Tanh, etc.

We normalized the $s_j$ scores using the softmax function.

$$n^j = Softmax(s_j) = \frac{\exp(s_j)}{\sum_{j=1}^{m} \exp(s_j)}, \tag{12}$$

The final output $o$ of BAGRU is obtained by multiplying the attention weight calculation and the feature vector:

$$o = \sum_{j=1}^{m} n^j \times f_j, \tag{13}$$

In BAGRU, the attention mechanism is used to learn the time step and feature importance and the essential features will be assigned higher weights, thereby improving gesture recognition performance.

### 4.5.3. Improved ResNet18

In recent years, CNN has become a research hotspot due to its good performance in areas such as image processing and speech recognition. ResNet is a high-performing variant of CNN, which is good at extracting local and spatial features of data, with excellent performance and fewer model parameters. However, the data features extracted by ResNet have the same weight. To solve this problem, we added the Convolutional Block Attention Module (CBAM) to ResNet18, which can adaptively focus on essential features in both channels and space [44]. The structure is shown in Figure 9.

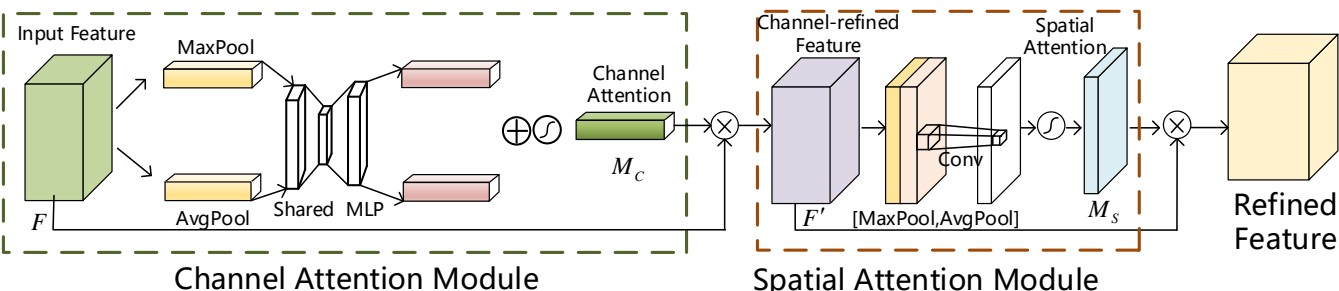

**Figure 9.** CBAM structure.

The channel attention is in front of the spatial attention. The input feature map performs maximum global pooling and global draw pooling, respectively. Then the max pooling and average pooling results are processed using two shared, fully connected layers. Then add the processing results and obtain the normalized attention weight through the Sigmoid function. Finally, the obtained weights are multiplied by the original input feature map, thus completing the redefinition of the original features by channel attention. The formula is shown below:

$$
\begin{aligned}
M_c(F) &= \sigma(MLP(AvgPool(F)) + MLP(MaxPool(F))) \\
&= \sigma\left(W_1\left(W_0\left(F_{avg}^c\right)\right) + W_1(W_0(F_{max}^c))\right)
\end{aligned} , \tag{14}
$$

$$
F' = M_C(F) \otimes F, \tag{15}
$$

The spatial attention mechanism will take maximum pooling and average pooling on the channel of each feature point of the input feature layer. After that, the two results are stacked. The feature map is then downscaled using a convolution with a channel number of 1, a convolution kernel of $1 \times 1$ and the Relu activation function. Then take a Sigmoid activation function. At this point, the weight of each feature point of the input feature layer is obtained. After obtaining the weight, we multiply this weight by the feature map obtained by channel attention and finally output a new feature map to complete the recalibration of the feature map in the two dimensions of space and channel. as shown in the formula below:

$$
\begin{aligned}
M_s(F') &= \sigma\left(f^{1\times1}([AvgPool(F); MaxPool(F)])\right) \\
&= \sigma\left(f^{1\times1}\left(\left[F_{avg}^s; F_{max}^s\right]\right)\right)
\end{aligned} , \tag{16}
$$

$$
F'' = M_S(F') \otimes F', \tag{17}
$$

With the emergence of neural network structures such as LeNet-5 and AlexNet, CNN has gradually developed from a neural network with a single convolution and pooling operation to a neural network with average pooling, maximum pooling, dropout and nonlinear functions. While the network structure has become increasingly complex, experts have found that the efficiency of the neural network has not achieved the expected effect. On the contrary, it is prone to gradient disappearance and so on. Therefore, the ResNet neural network appeared. It is a way of adding shortcuts between several convolutional layers to solve the problem of poor training when the number of neural network layers is gradually increased. We adopt ResNet18 because of its low complexity, few parameters and better performance. Our ResNet18 is structured by adding a CBAM after the first convolutional layer and a CBAM after the last residual block. The structure is shown in Figure 7. The ResNet18 residual block is shown in Figure 10.

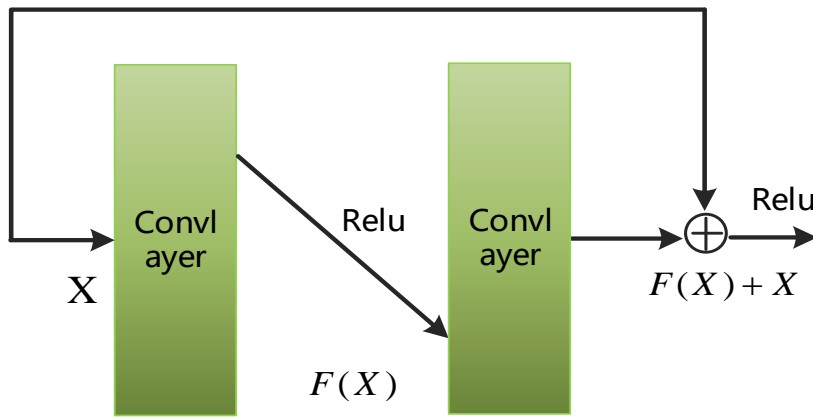

**Figure 10.** ResNet18 residual blocks.

The figure above shows a residual block of ResNet18, where $F(X)$ is the residual function, and the residual function we use is the Relu function. The advantage of the residual network is that the complete output learning problem is reduced to the residual learning problem. This structure's advantage is adding a shortcut outside the two convolutional layers, which not only solves the problem of gradient disappearance but also improves the computational efficiency. So $X$ can be output as $F(X) + X$ after passing through two convolutional layers.

A residual block can be represented as follows:

$$x_L = x_l + \sum_{i=1}^{L-1} F(x_i, w_i), \tag{18}$$

where $x_l$ is the input of the residual block and $F(x_i, w_i)$ is the value of $X$ learned through a convolutional layer. $L$ is the unit accumulation of individual residual block features and MLP is the accumulation of feature matrices, according to the chain rule of derivatives used in Back Propagation (BP). The gradient of the loss function $\varepsilon$ to $x_l$ can be expressed as:

$$\frac{dloss}{dxl} = \frac{dloss}{dxL} * \frac{dxL}{dxl} = \frac{dloss}{dxL}\left(1 + \frac{d\sum_{i=1}^{L-1} F(x_i, w_i)}{dxl}\right), \tag{19}$$

where 1 means that the residual block can inherit the gradient unconditionally. When $d\sum_{i=1}^{L-1} F(x_i, w_i)$ is close to 0, the model's gradient is still the gradient when the number of network layers is small, so the residual block can solve the problem that the model is challenging to train due to the increasing number of network layers.

## 5. Experimental Design and Analysis

To test the feasibility of the Wi-GC solution in a natural environment, we use the Intel 5300 network card resolution based on the 802.11n protocol. The experimental equipment is two laptops equipped with Intel 5300 network cards. One notebook is the transmitter (Tx) and the other is the receiver (Rx). Tx has one antenna, and Rx has three antennas. The antenna contacts of the transmitter and receiver are connected to a 1.5 m external antenna, as shown in Figure 11a. We select two real scenarios for experiments, as shown in Figure 11b,c.

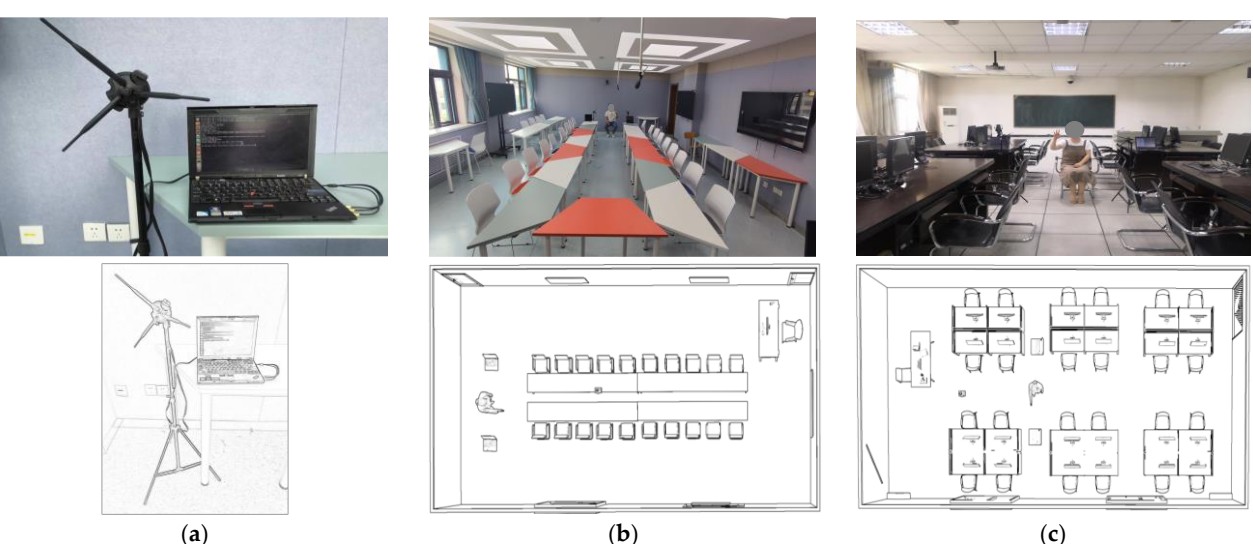

|     |     |     |
| :-: | :-: | :-: |
| (**a**) | (**b**) | (**c**) |

**Figure 11.** Experimental equipment and environment: (**a**) Experimental equipment. (**b**) Conference room environment. (**c**) Office environment.

We selected 8 experimenters aged 23–28 to collect CSI data. Each experimenter sat in a designated position and performed each gesture for 2 s. The training and test samples

ratio were 4:1. To make the experimental results more concise; we abbreviated OK, Arm sliding left, Arm sliding right, Push, Rotate, and Raise hand as OK, ASL, ASR, Push, RO, and RD, respectively.

### 5.1. Experimental Analysis

#### 5.1.1. Performance of Cross-Domain Gesture Recognition

Cross-domain refers to different environments. To verify the cross-domain performance of Wi-GC, we conducted experiments in conference rooms and office environments. We asked the experimenters to perform six gestures in the two environments. The experimental results are shown in Figure 12a. We set the equipment distance to 1 m, 2 m, 3 m, 4 m, 5 m and 6 m for experiments to find the optimal equipment distance. The results are shown in Figure 12b.

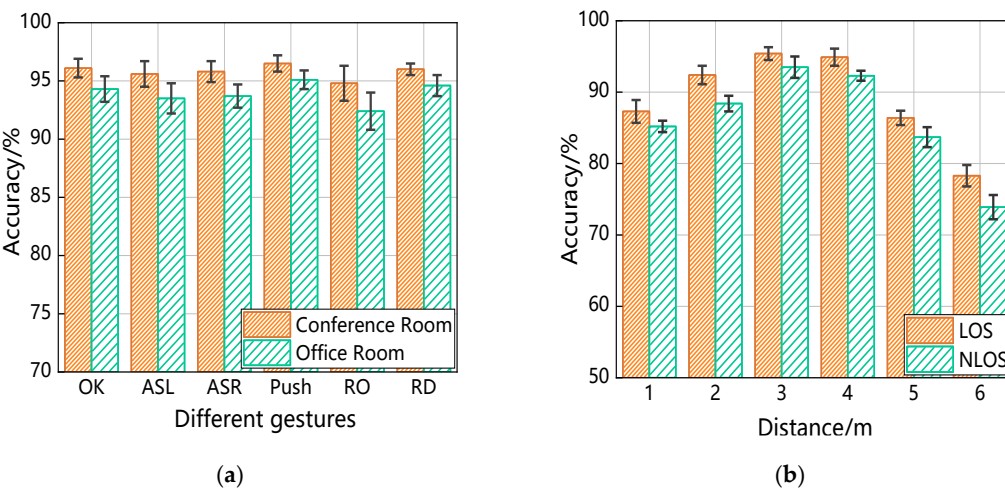

(**a**)                                                           (**b**)

**Figure 12.** The impact of cross-domain on gesture recognition: (**a**) The effect of different environments on gestures. (**b**) The effect of device distance on gestures.

As shown in Figure 12a, the accuracy rate of the conference room environment is higher than that of the office environment because the office environment is more complex and has more interference. From the final results, the average gesture recognition rate of both environmental domains is higher than 92%, which shows that Wi-GC has good recognition performance for cross-domain gestures. As can be seen from Figure 12b, no matter how far apart the two devices are, the recognition rate of LOS is always higher than that of NLOS. This is because the information received at the receiving end has not only the data affected by the action but also the influence of the items in the environment, resulting in lower recognition rates. When the distance between the transmitter and receiver is about 3m, the recognition rate is the highest, whether in a LOS environment or an NLOS environment. The signal propagation distance is short, the signal attenuation is small, and the perception range is extensive. When the device distance is equal to or greater than 5m, the gesture recognition rate decreases because the signal propagation distance is long, the signal attenuation increases and the perceived range is small.

#### 5.1.2. Effect of Cross-Targeting on Experimental Results

Cross-target refers to different users. To test the effect of making the same gesture across targets and making different gestures on the same mark on the recognition rate, we selected four other experimenters (two men and two women). Four experimenters were asked to perform the same gesture in the conference room. Then one of the experimenters was asked to complete all the gestures proposed to verify the user's influence on the Wi-GC method. The result is shown in Figure 13.

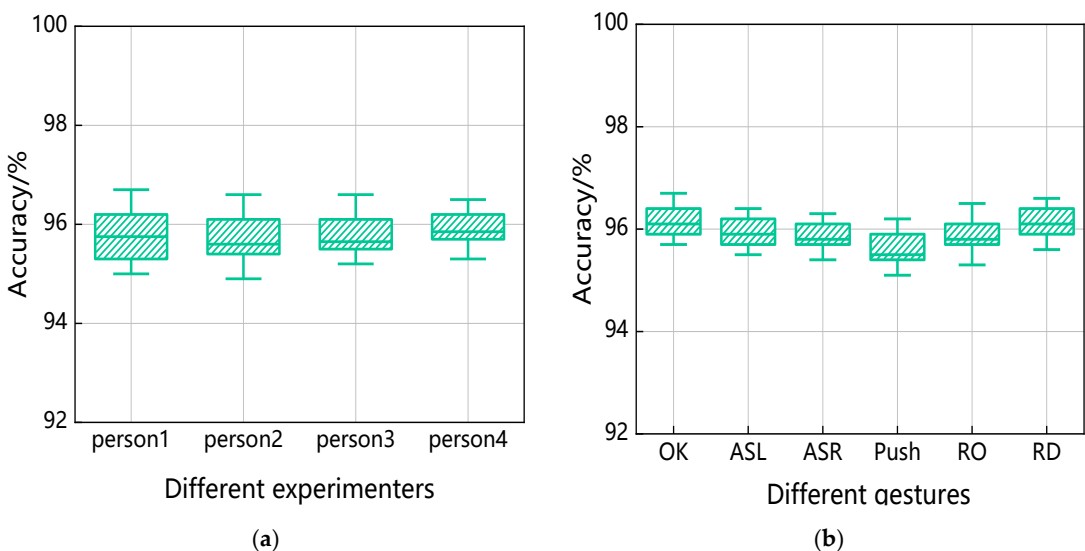

**Figure 13.** The effect of cross-targeting on gesture recognition: (**a**) Effect of different experimenters on gesture recognition. (**b**) Effect of the same experimenter on different gestures recognition.

Figure 13a shows that the gesture recognition rate of the four experimenters is more than 95%. It shows that changes in experimental personnel have little effect on the results. The data in Figure 13b shows that the recognition rate of each gesture achieved an accuracy rate of over 95.2%. This indicates that the same person's different gestures do not affect the experimental results. The above two experimental results verify that our proposed Wi-GC has high robustness.

5.1.3. Effect of Different Positions on Experimental Results

According to the characteristics of the CSI signal, it is sensitive to changes in position. In order to investigate the effect of location on recognition rate, the experiment was explicitly designed so that the experimenter's gesture direction was fixed and only the experimenter's location was changed, as shown in Figure 14. In our proposed method, sample data of six gestures were collected from each position and used to recognize sample data from each location. The final comparison results are shown in Figure 15.

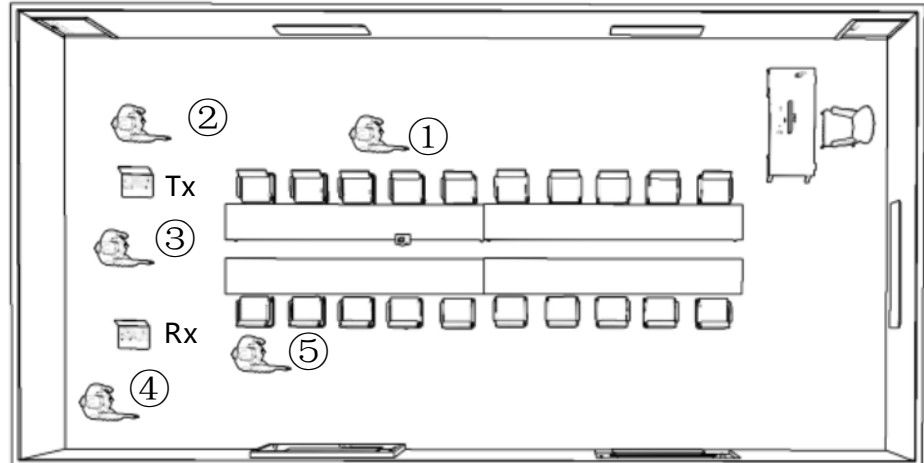

**Figure 14.** Experimental position design. The numbers in the diagram represent the position of the personnel.

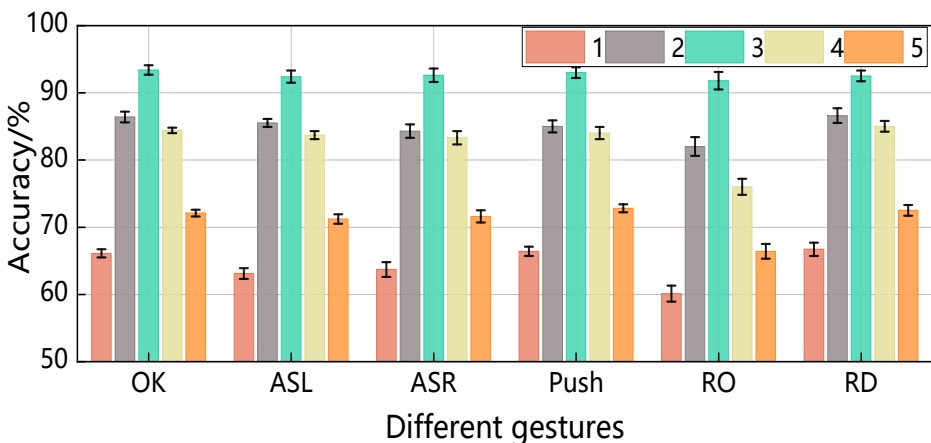

**Figure 15.** The effect of position on gesture accuracy.

As seen from the experimental results in Figure 15, position changes significantly impact the CSI signal, resulting in different levels of accuracy at each position. Location 1 has low accuracy because it is next to the conference table and the furthest away from the device, which creates more multi-path effects, resulting in the lowest accuracy. Position 3has the highest accuracy because it is closer to the transceiver and has fewer multi-path effects. Positions 4 and 2 are equally accurate, and both are affected by walls. Position 5 is more accurate than position 1 because it is above and relatively close to the receiving end, resulting in a slightly higher recognition rate.

### 5.2. Experimental Evaluation

#### 5.2.1. Algorithm Performance Comparison

The feature extraction algorithm is crucial in our model. Therefore, to explore the feasibility of our feature extraction RAGRU algorithm, it is compared with the traditional feature extraction algorithms CNN, BiGRU, and BiLSTM. The experimental results are shown in Figure 16a. Figure 16b shows the comparison results of each algorithm's gesture recognition rates since the number of hidden nodes in our BAGRU algorithm is an important parameter. To explore the optimal number of nodes, we chose different numbers of neurons and conducted comparison experiments in a conference room environment, and the results are shown in Figure 16c.

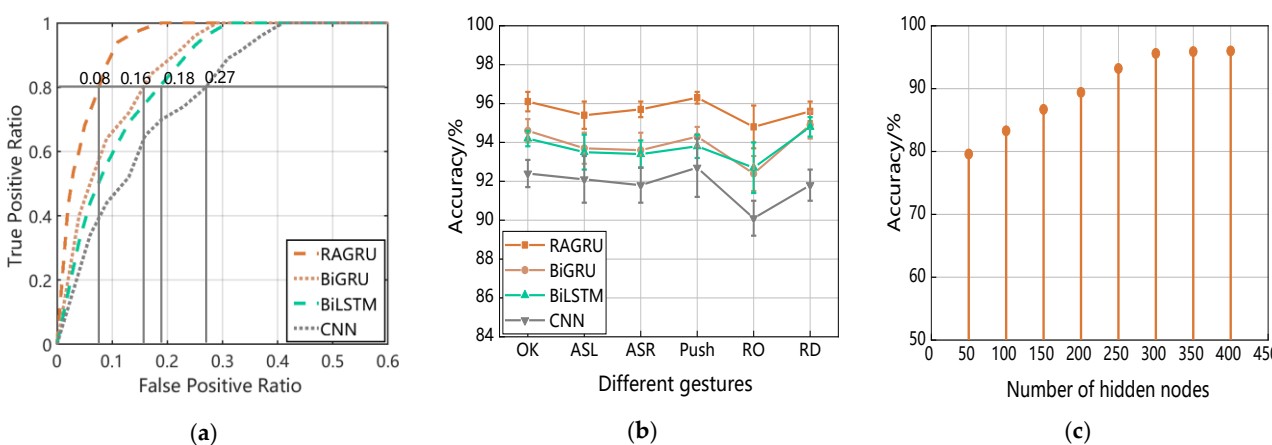

**Figure 16.** Algorithm performance comparison: (**a**) Comparison of feature extraction algorithms. (**b**) Impact of different feature extraction algorithms on gesture accuracy. (**c**) The effect of different numbers of hidden nodes on the accuracy of gestures.

Figure 16a shows different methods' Receiver Operator Curves (ROC). The x-axis represents the False Positives Rate (FPR), and the y-axis represents the True Positives Rate (TPR). When the RAGRU algorithm's true positive reaches 0.8, the false positive is only 0.08, which is the best. When the TPR of the BiGRU algorithm is 0.8, the FPR is 0.16, which is slightly lower than that of RAGRU. When the TPR of the BiGRU algorithm is 0.8, the FPR is 0.18, performance somewhat lower than BiGRU. When the TPR of the CNN algorithm is 0.8, the FPR is 0.27, and the performance is the worst. Combined above, RAGRU has the best performance because it not only extracts features from space and time but also introduces an attention mechanism, so it is better than BiGRU and BiLSTM. BiGRU and BiLSTM are about equal; BiGRU is a variant of BiLSTM, but the parameters are reduced, the complexity is low, and the performance is not much different from LSTM. Both consider past and future data information, so the performance is better than CNN. Figure 16b shows the gesture recognition results of the four algorithms, from which we can see that our proposed method has the best performance, so the RAGRU algorithm is feasible. From the results in Figure 16c, when the number of hidden nodes is 50, the gesture recognition rate is shallow, but when the number of hidden nodes increases from 50 to 300, the gesture accuracy rate reaches more than 95%. When the value of the hidden node rises again, the gesture recognition rate changes slightly. Since the number of hidden nodes increases, the training time will be longer since more hidden nodes lead to longer training time. Therefore, our BAGRU chooses 300 hidden nodes for gesture recognition.

### 5.2.2. The Impact of Dynamic Environments

In a natural environment, the indoor area will inevitably introduce dynamic changes in the environment, such as changes in the positions of tables and chairs and the addition of objects. These changes can cause the trained model to fail to recognize gestures because Wi-Fi signals are particularly sensitive to environmental changes. We designed two additional sets of experiments to examine the robustness of our Wi-GC method. We designed two environments based on the original layout of the meeting room. Program 1: we changed the structure of the conference room environment, as shown in Figure 17a, and added some static obstacles, including coffee tables, plants and chairs, to the test area. Program 2: when laboratory personnel are doing experiments at the designated location, we arranged for another experimental user to walk back and forth on the other side of the conference table, as shown in Figure 17b. The experimental results of the two sets of environments are shown in Figure 18.

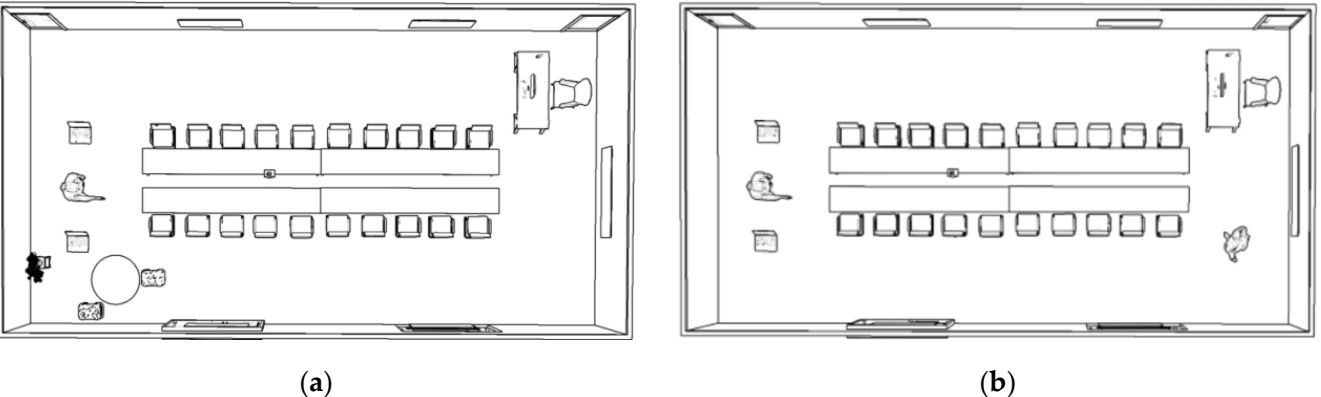

(**a**)　　　　　　　　　　　　　　　　　　　　　　　　　　　(**b**)

**Figure 17.** Environmental design: (**a**) Static environment. (**b**) Dynamic environment.

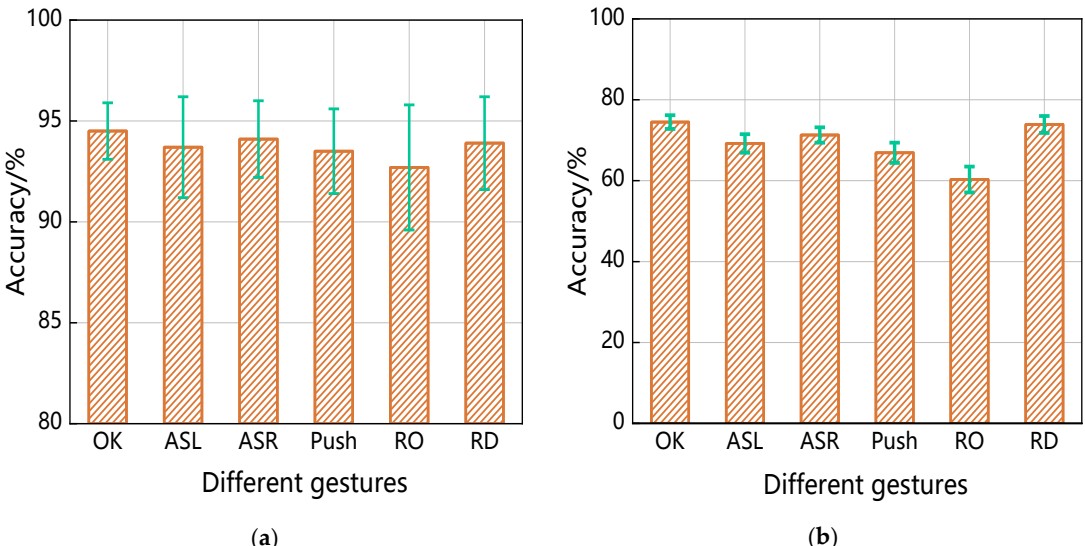

**Figure 18.** The influence of the environment on gestures: (**a**) The effect of static environments on gestures. (**b**) The effect of dynamic environments on gestures.

As seen from the results in Figure 18a,b, the accuracy of both schemes decreases as the environment changes. In program 1:, the accuracy of Wi-GC only drops by about 2%, indicating that static environments have little impact on the Wi-GC method. The second solution is considerably less accurate, as environmental changes significantly impact the Wi-Fi signal, but the accuracy of each gesture is above 60%. The results show that the Wi-GC method is robust and can adapt to static environmental disturbances.

### 5.2.3. Method Generalisation Ability Test

To test the generalization ability of our method, we design two sets of experiments. In the first group, in the conference room environment, we used the data of the first three experimenters for training and the fourth experimenter for testing. The results are shown in Figure 19a; the second group in the office in the environment, we use the same experimental design as the conference, and the experimental results are shown in Figure 19b.

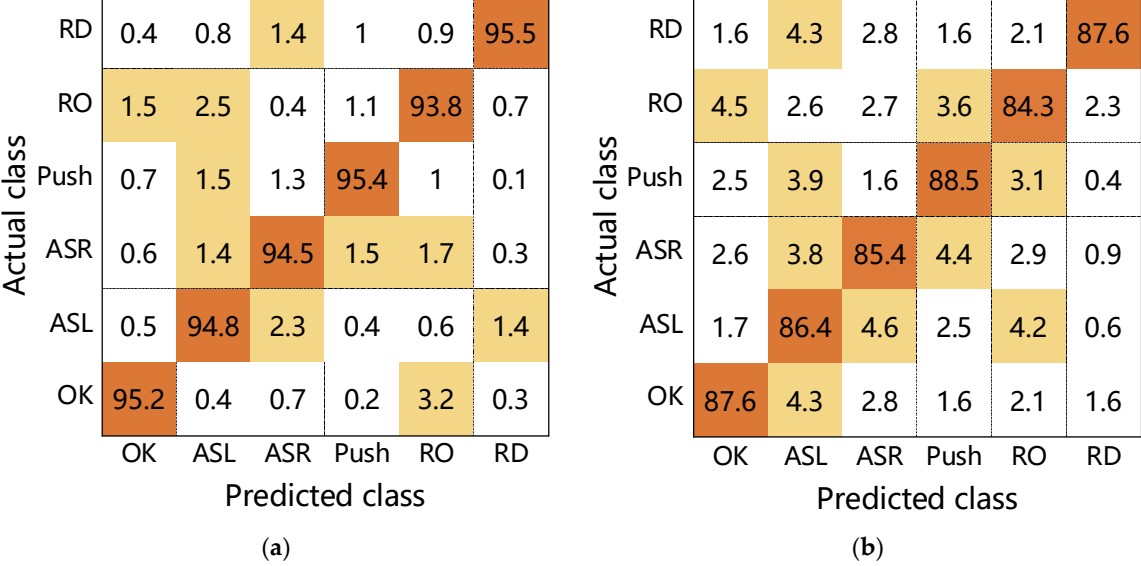

**Figure 19.** System generalisation capability test: (**a**) Testing in a conference room environment. (**b**) Testing in an office environment.

From the results in Figure 19a, the omitted experimenter data when tested can be seen. The accuracy of each gesture is above 93%. This is because the gesture data we train are under guidance. As long as The tester makes the gesture specification and the Wi-GC method will have higher accuracy. As can be seen from the experimental results in Figure 19b, the accuracy rate for the missing personnel experiment in the office environment is 84%. Because the office environment is more complex and has more distractions than the boardroom environment, the accuracy rate is lower.

### 5.2.4. Comparison with Other Existing Methods

To verify the performance of the Wi-GC method. Our method is compared with the Ref. [28], CrossGR [45] and WiNum [46] in a conference room environment. The CrossGR method first uses CNN to extract features from CSI gesture data and then uses machine learning to classify gesture-related features. The WiNum method uses discrete wavelet transform to denoise CSI data, the AGS algorithm adaptively segments gestures, and the Gradient Boosting Decision Tree (GBDT) integrates learning algorithm recognizes gestures. The comparison results are shown in Table 1.

**Table 1.** Results of Wi-GC versus other methods.

| Environment | Method | OK | ASL | ASR | Push | RO | RD |
|---|---|---|---|---|---|---|---|
| Conference Room | Ref. [28] | 89.7% | 88.5% | 88.2% | 90.5% | 87.4% | 89.9% |
| | CrossGR | 89.5% | 89.1% | 88.6% | 90.2% | 86.9% | 90.2% |
| | WiNum | 87.5% | 87.1% | 86.7% | 88.3% | 85.4% | 88.1% |
| | Wi-GC | 96.8% | 95.4% | 95.7% | 96.5% | 95.1% | 96.3% |

As seen in Table 1, the WiNum method has the worst performance because it does not perform feature extraction on CSI data but feeds the segmented data directly into GBDT for classification. Thus, it ignores the essential features of CSI information and does not perform as well as the other methods. The performance of Ref. [28] and the CrossGR method are comparable and CNN is used for feature extraction. CNN mainly focuses on the local spatial features of the data, not on the time series features. However, our method extracts the critical features of gestures from space and time, so the performance of other methods is inferior to Wi-GC.

### 5.2.5. Method Performance Evaluation

To evaluate the performance of the Wi-GC method, we conducted experiments on the Widar3.0 dataset [47], the CSIDA dataset [48] and our data set. Widar3.0 is composed of two sub-datasets, contributed by researchers at Tsinghua University, and has six gestures: push-pull, sweep, clap, slide, circle, and zigzag. CSIDA is a CSI-based gesture data set with six gestures: up, down, left, right, circle, and zigzag, proposed by Zhang et al. Experimental data are from the office environment; the results are shown in Figure 20.

As can be seen from Figure 20, the Widar 3.0 dataset has the highest accuracy rate, with an average accuracy of 93.7%. This is because it is a relatively simple office environment, which produces fewer multi-path effects. The difference in accuracy between the CSIDA dataset and our dataset is small, due to their similar office layouts, with the perception area in the middle of two rows of desks. The environment is more complex than the Widar 3.0 dataset and, thus, slightly less accurate. Overall, the average accuracy of all three datasets was above 91%, indicating the excellent performance of our method.

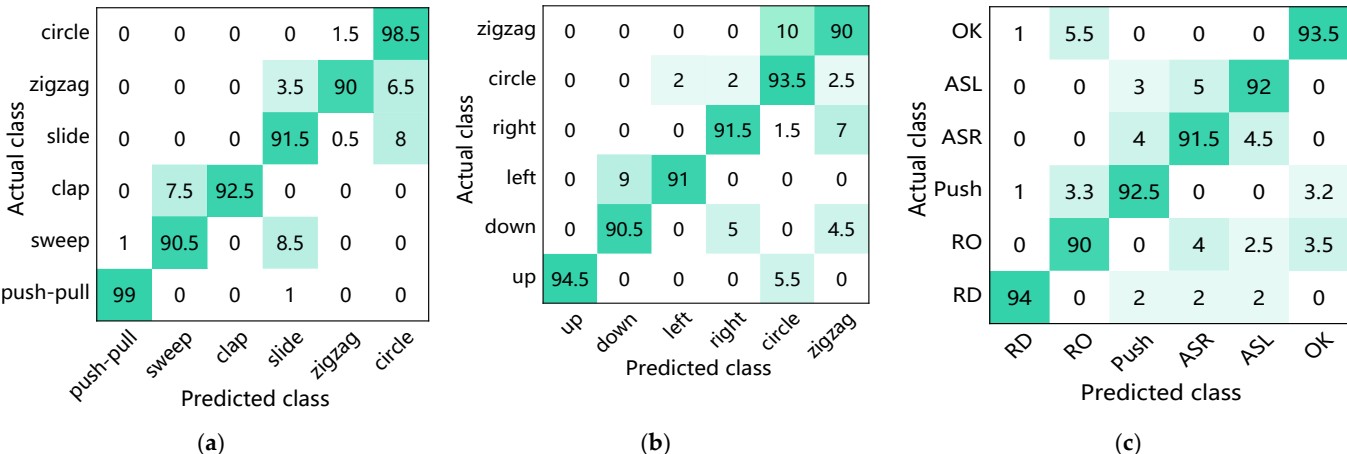

**Figure 20.** Method performance evaluation: (**a**) Widar3.0 data set validation. (**b**) CSIDA data set validation. (**c**) our data set validation.

5.2.6. Analysis of Method Time

To explore the time-consuming nature of our method, Wi-GC was compared with Ref. [28], CrossGR [45] and WiNum [46]. In the experiment, the sample data of a gesture is used and the training and test samples were in the ratio of 4:1. The comparison results are shown in Table 2.

**Table 2.** Analysis of method time.

| Method | Training Time of One Sample (s) | Testing Time of One Sample (s) | Mean Accuracy (%) |
|---|---|---|---|
| Ref. [28] | 0.09374 | 0.00617 | 89.9 |
| CrossGR | 0.08168 | 0.00586 | 90.3 |
| WiNum | 0.05263 | 0.00317 | 87.2 |
| Wi-GC (ours) | 0.23815 | 0.05618 | 94.7 |

As can be seen from Table 2, WiNum takes the least time, because the algorithms it uses are all machine learning and there is no feature extraction component. Ref. [28] and CrossGR have similar feature extraction and classification algorithms, except that Ref. [28] is slightly more complex than CrossGR in data processing, so it takes more time. The Wi-GC method takes the most time because its algorithm structure is more complicated than all the other methods, but the time taken is within acceptable limits.

## 6. Conclusions

The method first collects a large amount of sample gesture data and selects antennas sensitive to gesture movements, then utilize a combination of Kalman and wavelet filters to noise reduce the selected antenna data, followed by a segmentation algorithm using time series differences to find the start and end points of the gesture data. Since the CSI data is temporal and its amplitude is spatial, we use BAGRU to extract the temporal features of the gesture and RNet18 to extract the spatial characteristics. Finally, we fuse the two features input into softmax for classification. In the experiment, we analyzed the influence of different factors on gesture recognition and evaluated the performance of Wi-GC in two different scenarios. The experimental results show that the average accuracy of gesture actions is above 92%. Therefore, Wi-GC can be used as a solution for emerging IoT applications. Furthermore, we will improve the method's performance in future work and consider studying problems across multiple environments.

**Author Contributions:** Conceptualization, X.D., Y.B. and Z.H.; methodology, X.D. and Y.B.; software, Y.B. and G.L.; validation, X.D., Y.B. and Z.H.; formal analysis, X.D., Y.B. and G.L.; investigation, Z.H.; resources, X.D. and Y.B.; data curation, X.D., Y.B. and G.L.; writing—original draft preparation, Y.B. and Z.H.; writing—review and editing, X.D., Y.B. and G.L.; visualization, X.D. and Z.H.; supervision, X.D.; project administration, X.D. and Z.H.; funding acquisition, X.D. and Z.H. All authors have read and agreed to the published version of the manuscript.

**Funding:** This work was supported by the National Natural Science Foundation of China (Grant 62162056, 62262061), Industrial Support Foundations of Gansu (Grant No. 2021CYZC-06), 2020 Lanzhou City Talent Innovation and Entrepreneurship Project (2020-RC-116), and Gansu Provincial Department of Education: Industry Support Program Project (2022CYZC-12).

**Institutional Review Board Statement:** The study does not require ethical approval.

**Informed Consent Statement:** Written informed consent was obtained from the experimentalists for this paper.

**Data Availability Statement:** The study did not report any data.

**Conflicts of Interest:** The authors declare no conflict of interest.

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
