# Peer review of "Wi-GC: A Deep Spatiotemporal Gesture Recognition Method Based on Wi-Fi Signal"

_applsci, doi:10.3390/app122010425_

Round 1

Reviewer 1 Report

·      What are these initials RAGRU, CSI, Wi-GC. It should be full name followed by initials when firstly used.

·      Paper have lot of typos for example at pg no 2, line 47, page no 7 line 264.

·      In the literature section paper citation is not in appropriate. Always use author name with reference or write according to format of the journal.

·      Introduction section is not very clear. What is objective of this study? What is the motivation behind it and what is the problem in the existing study?

·      What is the role of attention mechanism in GRU?

·      Why only BGRU and RNET18. Why not some other model?

·      IN the literature recent work is missing.

·      What is the application of wifi signal based gesture recognition? What is the useful ness of such system?

·      What is the time analysis of the proposed approach? Is it feasible in real-time with how much accuracy?

·      Author should show experiments on more dataset? Result is not convincing and also comparison with literature is missing?

·      Paper has lot of English grammar mistake. Proof read is needed.

Author Response

Hello!

Thank you very much for your interest in our research team, as well as for your very rigorous and professional review and guidance of the manuscript of the Applied Science - 1952518.

Revision of the manuscript of the Applied Sciences - 1952518. Completed. The manuscript is revised in detail according to the revision opinions of the manuscript reviewer and the editing teacher, and the changes are marked in our manuscript using the "Track Changes" function. We have already marked the changes in red font in the modification instructions. We hereby send you a draft revision (with or without "Track Changes") and a description of the changes. If you find any problems, please contact us in time.

I wish you good health, smooth work and happy family!

Kind regards

Manuscripts of all authors of Applied Science-1952518.

Reviewer 1

What are these initials RAGRU, CSI, Wi-GC. It should be full name followed by initials when firstly used.

Reply:

Thank you very much for your suggestion. We have modified the abbreviations RAGRU, Wi-GC, and CSI in the abstract, with RAGRU being the name of our proposed model. It was amended to read, "The segmented gesture data is fed into our proposed RAGRU model". Wi-GC is short for the name of our paper. It was amended to read, "we have designed a deep spatiotemporal gesture recognition method based on Wi-Fi signals, namely Wi-GC". CSI was amended to read "Channel State Information (CSI)".

  1. Paper have lot of typos for example at pg no 2, line 47, page no 7 line 264.

Reply:

Thank you very much for pointing out this error, which we have carefully corrected in the manuscript.

  1. In the literature section paper citation is not in appropriate. Always use author name with reference or write according to format of the journal.

Reply:

Thank you very much for your suggestion., We have revised the Literature [14] to read "Thariq et al. [14]", Literature [15] was amended to read "Zhang et al. [15]", Literature [16] was amended to read "Fang et al. [16]", Literature [5] was amended to read "Zhu et al. [5]", Literature [17] was amended to read "Nguyen-Trong et al. [17]", Literature [18] was amended to read "Lv et al. [18]",Literature [19] was amended to read "Jung et al. [19]", Literature [20] was amended to read "Colli Alfaro et al. [20]", Literature [21] was amended to read "Cheng et al. [21]", Literature [6] was amended to read "Nair et al. [6]", Literature [7] was amended to read "Zhang et al. [7]", Literature [22] was amended to read "Hao et al. [22]", Literature [23] was amended to read "Dang et al. [23]", Literature [24] was amended to read "Han et al. [24]", Literature [25] was amended to read "Yang et al. [25]", literature [31] was amended to read "Ref. [31]", literature [33] was amended to read "Ref. [33]", literature [36] was amended to read "Ref. [36]", literature [37] was amended to read "Ref. [37]".

  1. Introduction section is not very clear. What is objective of this study? What is the motivation behind it and what is the problem in the existing study?

Reply:

Thank you very much for your suggestion. We have amended this in the first paragraph of the introduction to the revised draft. The changes are shown below:

Gesture recognition plays a vital role research field of Human-Computer Interaction (HCI)[1]. It supports many emerging Internet of Things (IoT) applications such as user recognition [2], smart home [3], healthcare [4] etc. Generally, the technologies based on gesture recognition mainly include sensors [5], web cameras [6], and millimetre-wave radars [7]. However, they all have certain limitations. For example, the sensor will cause an additional body burden to the user, and its deployment and maintenance costs are high. The camera will expose the user's privacy and dead spots in the shot. Millimetre-wave radar signal attenuation is significant, and the price is high. Recently, Wi-Fi-based gesture recognition methods [8] have become a hot research topic with non-contact, easy deployment, security and low-cost advantages. However, most of the current Wi-Fi-based gesture recognition methods extract the temporal features of gestures and ignore the spatial features, which affects the accuracy of gesture recognition to different degrees. To this end, we propose a method to obtain both temporal and spatial characteristics of gestures, thus improving gesture recognition accuracy in Wi-Fi-based environments.

  1. What is objective of this study?

Reply:

Thank you very much for your question. The objectives of this research are: Gesture recognition plays a vital role research field of Human-Computer Interaction (HCI). It supports many emerging Internet of Things (IoT) applications such as user identification, smart homes, healthcare etc.

  1. What is the motivation behind it?

Reply:

Thank you very much for your question. The motivation behind it is: Most current Wi-Fi-based gesture recognition methods extract the temporal features of the gesture and ignore the spatial features, providing insufficient recognition accuracy.

  1. what is the problem in the existing study?

Reply:

Thank you very much for your question. The issues in existing research are: Existing research technologies have limitations, such as sensors that place an additional physical burden on the user and it is expensive to deploy and maintain; cameras that expose the user's privacy and have dead spots to capture; and millimetre wave radar with high signal attenuation and high cost.

  1. What is the role of attention mechanism in GRU?

Reply:

Thank you very much for your question. The role of the attention mechanism introduced in GRU is that features learned in traditional GRU networks have the same weight in gesture classification. To solve this problem, we have introduced an attention mechanism in GRU that assigns a weight to each time step and feature. And can learn the importance of each time step and feature autonomously. Higher weights are then transferred to the more critical time steps and characteristics, thus improving the method's performance.

  1. Why only BGRU and RNET18. Why not some other model?

Reply:

Thank you very much for your question. We only have BAGRU and RNET18 in our method and no other models, and this is because of how this paper has achieved the expected results according to the experimental results. Thank you again for your comments. We will then continue to investigate if other methods will obtain better results.

  1. IN the literature recent work is missing.

Thank you very much for your suggestion. We did miss some recent and relevant work. Therefore, we have added the following references.

Line 188: We have added content: Shi et al. [26] propose a gesture recognition method based on Wi-Fi signals. The amplitude and phase of the CSI are first extracted from the Wi-Fi signal, the amplitude is filtered, and the phase is expanded and linearly transformed. The CSI signal is then converted to a Red Green Blue (RGB) image using normalization and interpolation. Finally, the combined amplitude and phase RGB images are classified using a lightweight deep network model based on MobileNet_V2. Wang et al. [27] propose a gesture recognition system based on a matched average joint learning framework (WiMA). The system uses parameter-matching collaborative learning to train a gesture prediction model. Static bias in the data is first removed using CSI amplitude conjugate multiplication, and noise and random bias are removed using a band-pass filter. The data's Doppler Frequency Shift (DFS) spectrum is extracted, and a Body-coordinate Velocity Profile (BVP) is generated. The features of the BVP are then extracted using CNN-LSTM, and finally, the gesture features are classified using softmax.WiFi-Based Low-Complexity Gesture Recognition Using Categorization. Kim et al. [28] propose a low-complexity gesture classification recognition based on Wi-Fi. The CSI data is first processed using different techniques, and the pre-processed data is segmented. Then features are extracted using a deep degree learning model, and finally, the extracted features are classified using SVM. Ding et al. [29] proposed a gesture recognition scheme based on the multimodal fusion Gaussian mixture model (GMM). Their proposed GDS algorithm was used to segment the gesture data and then use Singular Value Decomposition (SVD) to derive multi-view features from CSI measurements on all subcarriers of the Wi-Fi receiver to represent gesture features. Using Multimodal Factorized Bilinear (MFB) pooling method to efficiently fusion features from all receiver antennas and finally identify various gestures by integrating multimodal fusion and GMM.

We have added references [26][27][28][29] as follows:

  1. Shi, L.; Wang, Y.; Qian, H. CSI gesture recognition method based on lightweight deep network. Third International Conference on Artificial Intelligence and Electromechanical Automation (AIEA 2022). Changsha, China, 7 September 2022; 12329, pp.241-246.
  2. Wang, Z.; Zhang, W.; Wu, X.; Wang, X. Matched Averaging Federated Learning Gesture Recognition with WiFi Signals. 2021 7th International Conference on Big Data Computing and Communications (BigCom). Deqing, China, 13-15 August 2021; pp.38-43.
  3. Kim, J. S.; Jeon, W. S.; Jeong, D. G. WiFi-Based Low-Complexity Gesture Recognition Using Categorization. In 2022 IEEE 95th Vehicular Technology Conference (VTC2022-Spring). Helsinki, Finland, 19-22 June 2022; pp. 1-7.
  4. Ding, J.; Wang, Y.; Si, H.; Ma, J.; He, J.; Liang, K.; Fu, S. Multimodal Fusion-GMM based Gesture Recognition for Smart Home by WiFi Sensing. In 2022 IEEE 95th Vehicular Technology Conference (VTC2022-Spring). Helsinki, Finland, 19-22 June 2022; pp. 1-6.
  5. What is the application of WiFi signal based gesture recognition? What is the usefulness of such system?

Reply:

Thank you very much for your question. Gesture recognition based on Wi-Fi signals can be applied to smart homes, user identification, and healthcare. Its usefulness is that it is more convenient to operate and enables interaction with machines in a non-contact situation, avoiding the risk of germs spreading in public environments (e.g. hospitals, cinemas, etc.).

  1. What is the time analysis of the proposed approach? Is it feasible in real-time with how much accuracy?
  2. What is the time analysis of the proposed approach?

Reply:

Thank you very much for your suggestion. In response to our question about the time analysis of the proposed method, we did overlook the problem with the timing of the Wi-GC way, so we have added subsection 5.3.6 Experiments to the manuscript.

Line 674: We have added content:

5.3.6. Analysis of method time

To explore the time-consuming nature of our method, Wi-GC was compared with Ref. [25], CrossGR [38] and WiNum [39]. In the experiment, the sample data of a gesture is used. And the training and test samples were in the ratio of 4:1. The comparison results are shown in Table 2.

Table 2. Analysis of method time

Method

Training Time of One Sample (s)

Testing Time

of One Sample (s)

Mean Accuracy(%)

Ref. [25]

0.09374

0.00617

89.9

CrossGR

0.08168

0.00586

90.3

WiNum

0.05263

0.00317

87.2

Wi-GC(ours)

0.23815

0.05618

94.7

As can be seen from Table 2, WiNum takes the least time. Because the algorithms it uses are all machine learning, and there is no feature extraction component. Ref. [25] and CrossGR have similar feature extraction and classification algorithms, except that Ref. [25] is slightly more complex than CrossGR in data processing, so it takes more time. The Wi-GC method takes the most time because its algorithm structure is more complicated than all the other methods, but the time taken is within acceptable limits.

  1. Is it feasible in real-time with how much accuracy?

Reply:

Thank you very much for your question—the problem of whether the method can be real-time and the accuracy of real-time. We are also working on researching how to do this in real-time due to the limitations of current technology. It is something we will be working on in the future.

  1. Author should show experiments on more dataset? Result is not convincing and also comparison with literature is missing?
  2. Author should show experiments on more dataset?

Reply:

Thank you very much for your suggestion. We did overlook this issue for the experimental question that the author should show more datasets. Therefore, to test the performance of our method, we have added subsection 5.3.5 experiments to the manuscript.

Line 656: We have added content:

5.3.5. Method performance evaluation

To evaluate the performance of the Wi-GC method, we conducted experiments on the Widar3.0 dataset [44], the CSIDA dataset [45], and our data set. Widar3.0 is composed of two sub-datasets, contributed by researchers at Tsinghua University, and has six gestures: push-pull, sweep, clap, slide, circle, and zigzag. CSIDA is a CSI-based gesture data set with six gestures: up, down, left, right, circle, and zigzag, proposed by Zhang et al. Experimental data are from the office environment; the results are shown in Figure 20.

(a)

(b)

(c)

Figure 20. Method performance evaluation: (a) Widar3.0 data set validation. (b) CSIDA data  set validation. (c) our data set validation

As can be seen from Figure 20, the Widar 3.0 dataset has the highest accuracy rate, with an average accuracy of 93.7%. It is because it is a relatively simple office environment, which produces fewer multi-path effects. The difference in accuracy between the CSIDA dataset and our dataset is small, due to their similar office layouts, with the perception area in the middle of two rows of desks. The environment is more complex than the Widar 3.0 dataset and, thus, slightly less accurate. Overall, the average accuracy of all three datasets was above 91%, indicating the excellent performance of our method.

  1. Result is not convincing and also comparison with literature is missing?

Reply:

Thank you very much for your suggestion regarding the lack of comparison with the literature. We have compared the Wi-GC method with the literature [25], CrossGR [38], and WiNum [39] in a conference room environment. The experiment is set in section 5.3.4 of the manuscript.

5.3.4 Comparison with other existing methods

In order to verify the effectiveness of our Wi-GC method, our method is compared with the Ref. [25], CrossGR [42], and WiNum [43] in a conference room environment. CrossGR method first uses CNN to extract features from CSI gesture data and then uses machine learning to classify gesture-related features. WiNum method uses discrete wavelet transform to denoise CSI data, the AGS algorithm adaptively segments gestures, and Gradient Boosting Decision Tree (GBDT) integrated learning algorithm recognizes gestures. The comparison results are shown in Table 1:

Table 1. Results of Wi-GC versus other methods

Environment

Method

OK

ASL

ASR

Push

RO

RD

Conference Room

Ref. [25]

89.7%

88.5%

88.2%

90.5%

87.4%

89.9%

CrossGR

89.5%

89.1%

88.6%

90.2%

86.9%

90.2%

WiNum

87.5%

87.1%

86.7%

88.3%

85.4%

88.1%

Wi-GC

96.8%

95.4%

95.7%

96.5%

95.1%

96.3%

As seen from Table 1, the WiNum method has the worst performance. This is because it does not perform feature extraction on the CSI data but directly feeds the segmented data into GBDT for classification, thus ignoring the essential features of CSI information, and therefore does not perform as well as the other methods. The performance of the Ref. [25] and the CrossGR way are comparable, and CNN is used for feature extraction. CNN mainly focuses on the local spatial features of the data, not on the time series features. However, our method extracts the critical features of gestures from space and time, so the performance of other methods is inferior to Wi-GC.

  1. Paper has lot of English grammar mistake. Proof read is needed.

Reply:

Thank you very much for your suggestion. We are sorry that we had such a problem with our submitted manuscript. We have double-checked the manuscript and corrected the error.

In addition, in order to further improve the quality of the article, we re-checked the article based on experts' and editors' revision opinions. Inappropriate places in the article were edited and corrected, and the corrected content has also been marked in the revised draft.

Finally, all authors of manuscript Applied Sciences-1952518 would like to express their heartfelt thanks to the editorial teachers for their valuable suggestions. At the same time, I wish the expert teacher good health and all the best!

Reviewer 2 Report

The paper is very well written. The design and conducted experiments are clearly presented. The results in section 6 are useful and interesting, especially because multiple aspects are observed.

Minor remarks are:

- quality of figure 8 should be improved (also xt is missing in figure)

- correct few typos and grammar errors

- sentence after the Table I: "As seen from Table 1, no matter what method, the recognition rate of the office is lower than that of the conference room environment because the multipath effect of the office is more serious, resulting in a low accuracy rate." The results in Table I are for conference room, and there are no results for office environment in Table I, thus this sentence is confusing. Correct the meaning of this sentence.

Author Response

Dear expert teacher,

Hello!

Thank you very much for your care of our research group, and also for your very strict and professional review and guidance of manuscript Applied Sciences-1952518.

The revision work of manuscript Applied Sciences-1952518. has been completed. The manuscript has been revised in detail according to the revision opinions of manuscript reviewers and editor teachers, and the revisions have been marked up using the “Track Changes” function in our manuscript. And we have been marked the modification in red font in the modification instructions. We hereby send you the revised draft (with and without “Track Changes”) and the modification instructions. If you find any problems, please contact us in time.

I wish you good health, smooth work and a happy family!

Kind regards,

All authors of manuscript Applied Sciences-1952518.

Reviewer 2

Revision notes and responses to review comments:

The paper is very well written. The design and conducted experiments are clearly presented. The results in section 6 are useful and interesting, especially because multiple aspects are observed.

Minor remarks are:

  1. quality of figure 8 should be improved (also is missing in figure)

Reply:

Thank you very much for your suggestion. We have carefully revised Figure 8, and we thank you again.

  1. correct few typos and grammar errors

Reply:

Thank you very much for your suggestion. We are sorry that we had such a problem with our submitted manuscript. We have double-checked the manuscript and corrected the error.

  1. sentence after the Table I: "As seen from Table 1, no matter what method, the recognition rate of the office is lower than that of the conference room environment because the multipath effect of the office is more serious, resulting in a low accuracy rate." The results in Table I are for conference room, and there are no results for office environment in Table I, thus this sentence is confusing. Correct the meaning of this sentence.

Reply:

Thank you very much for your advice. We are very sorry. We had such a problem with the manuscript we submitted, and we have carefully revised the analysis in Table 1, and the changes are shown below.

As seen in Table 1, the WiNum method has the worst performance because it does not perform feature extraction on CSI data but feeds the segmented data directly into GBDT for classification. Thus, it ignores the essential features of CSI information and does not perform as well as the other methods. The performance of Ref. [25] and the CrossGR way are comparable, and CNN is used for feature extraction. CNN mainly focuses on the local spatial features of the data, not on the time series features. However, our method extracts the critical features of gestures from space and time, so the performance of other methods is inferior to Wi-GC.

In addition, in order to further improve the quality of the article, we re-checked the article based on experts' and editors' revision opinions. Inappropriate places in the article were edited and corrected, and the corrected content has also been marked in the revised draft.

Finally, all authors of manuscript Applied Sciences-1952518 would like to express their heartfelt thanks to the editorial teachers for their valuable suggestions. At the same time, I wish the expert teacher good health and all the best!

Round 2

Reviewer 1 Report

Author's have incorporated all the reviewer comments and paper quality is improved now.

Author should include following recent studies in the paper .

1) Compact joints encoding for skeleton-based dynamic hand gesture recognition

2)  A two stream convolutional neural network with bi-directional GRU model to classify dynamic hand gesture

3) 3D Skeletal Joints-Based Hand Gesture Spotting and Classification

Author Response

Dear expert teacher,

Hello!

Thank you very much for your care of our research group, and also for your very strict and professional review and guidance of manuscript Applied Sciences-1952518.

The revision work of manuscript Applied Sciences-1952518. has been completed. The manuscript has been revised in detail according to the revision opinions of manuscript reviewers and editor teachers, and the revisions have been marked up using the “Track Changes” function in our manuscript. And we have been marked the modification in red font in the modification instructions. We hereby send you the revised draft (with and without “Track Changes”) and the modification instructions. If you find any problems, please contact us in time.

I wish you good health, smooth work and a happy family!

Kind regards,

All authors of manuscript Applied Sciences-1952518.

Reviewer 1

Revision notes and responses to review comments:

  1. Author should include following recent studies in the paper .

Reply:

Thank you very much for your suggestion. We have added your suggested reference, which reads as follows:

Li et al. [22] present a novel skeleton-based dynamic gesture recognition framework. In a Spatially Perceptive stream (SP-stream), the compact joints are adaptively selected using their designed compact joint coding method for convex packages of the hand skeleton. And encode them as skeleton images to fully extract spatial features. In addition, they provide a Global Enhancement Module (GEM) to enhance the critical feature maps. In the temporal perception stream (TP-stream), they propose a Motion Perception Module (MPM) to strengthen the significant motion of the gesture on the X/Y/Z axes. Then the Feature Aggregation Module (FAM) is used to aggregate more time dynamics. Finally, the scores obtained from the spatial-aware and time-aware streams are averaged to get the final classification results. Verma et al. [23] proposed a hybrid deep-learning framework to recognize dynamic gestures. The features of each frame in the video need to be extracted to obtain temporal and dynamic information about the gestures made. So GoogleNet is used to extract the gesture features from the video. Finally, the extracted features are transferred to a Bidirectional Gated Recurrent Unit (BiGRU) network to classify gestures. Nguyen et al. [24] proposed a new continuous dynamic gesture recognition method. They use a gesture localization module to segment a video sequence of continuous gestures into individual gestures. Three residual 3D Convolution Neural Networks based on ResNet architectures (3D_ResNet) are used to extract the RGB、 optical flow and depth of the gestures features. Meanwhile, BiLSTM is used to extract the features of the 3D positions of the critical joints of the gestures. Finally, the weights of the fully connected layers are fused for gesture classification.

We have added references [22][23][24] as follows:

[22]  Li, Y.; Ma, D.; Yu, Y.; Wei, G.; Zhou, Y. Compact joints encoding for skeleton-based dynamic hand gesture recognition. Computers & Graphics. June 2021, 97:191-199.

[23]  Verma, B. A two stream convolutional neural network with bi-directional GRU model to classify dynamic hand gesture. Journal of Visual Communication and Image Representation. August 2022, 87:103554.

[24]  Nguyen, N. H.; Phan, T. D. T.; Kim, S. H.; Yang, H. J.; & Lee, G. S. 3D Skeletal Joints-Based Hand Gesture Spotting and Classification. Applied Sciences. 2021, 11(10):4689.

In addition, in order to further improve the quality of the article, we re-checked the article based on experts' and editors' revision opinions. Inappropriate places in the article were edited and corrected, and the corrected content has also been marked in the revised draft.

Finally, all authors of manuscript Applied Sciences-1952518 would like to express their heartfelt thanks to the editorial teachers for their valuable suggestions. At the same time, I wish the expert teacher good health and all the best!
